# TIAToolbox as an end-to-end library for advanced tissue image analytics

Johnathan Pocock [1,2], Simon Graham[1,2], Quoc Dang Vu[1], Mostafa Jahanifar[1], Srijay Deshpande[1], Giorgos Hadjigeorghiou[1], Adam Shephard [1], Raja Muhammad Saad Bashir[1], Mohsin Bilal [1], Wenqi Lu[1], David Epstein [1], Fayyaz Minhas [1], Nasir M. Rajpoot [1] & Shan E Ahmed Raza [1✉]

**Abstract**

**Background** Computational pathology has seen rapid growth in recent years, driven by advanced deep-learning algorithms. Due to the sheer size and complexity of multi-gigapixel whole-slide images, to the best of our knowledge, there is no open-source software library providing a generic end-to-end API for pathology image analysis using best practices. Most researchers have designed custom pipelines from the bottom up, restricting the development of advanced algorithms to specialist users. To help overcome this bottleneck, we present TIAToolbox, a Python toolbox designed to make computational pathology accessible to computational, biomedical, and clinical researchers.

**Methods** By creating modular and configurable components, we enable the implementation of computational pathology algorithms in a way that is easy to use, flexible and extensible. We consider common sub-tasks including reading whole slide image data, patch extraction, stain normalization and augmentation, model inference, and visualization. For each of these steps, we provide a user-friendly application programming interface for commonly used methods and models.

**Results** We demonstrate the use of the interface to construct a full computational pathology deep-learning pipeline. We show, with the help of examples, how state-of-the-art deep-learning algorithms can be reimplemented in a streamlined manner using our library with minimal effort.

**Conclusions** We provide a usable and adaptable library with efficient, cutting-edge, and unit-tested tools for data loading, pre-processing, model inference, post-processing, and visualization. This enables a range of users to easily build upon recent deep-learning developments in the computational pathology literature.

**Plain language summary**

Computational software is being introduced to pathology, the study of the causes and effects of disease. Recently various computational pathology algorithms have been developed to analyze digital histology images. However, the software code written for these algorithms often combines functionality from several software packages which have specific setup requirements and code styles. This makes it difficult to re-use this code in other projects. We developed a computational software named TIAToolbox to alleviate this problem and hope this will help accelerate the use of computational software in pathology.

---

[1] Tissue Image Analytics Centre, University of Warwick, Coventry, UK. [2] These authors contributed equally: Johnathan Pocock, Simon Graham.
✉email: shan.raza@warwick.ac.uk

Digitization of classical cellular pathology workflows through the deployment of digital whole slide image (WSI) scanners has resulted in significant progress in the development of computational pathology (CPath) image analysis techniques. Such advances have benefited greatly by adapting deep learning techniques from computer vision producing novel solutions to a variety of CPath problems, including nucleus instance segmentation[1], pathology image quality analysis[2] and WSI-level prediction[3,4]. Although many algorithms have been developed for the analysis of WSIs which all share the same basic components (such as WSI reading, patch extraction and feeding to deep neural networks), there is no single open-source generic library that unifies all the steps using best practice to process these images. Several published algorithms have their own packaged codebases which run in a task-specific environment, with tightly coupled interfaces, dependencies, and image format requirements. It is also common for there to be little to no code quality checks or unit testing. This may prevent code from a published peer-reviewed method from being able to run out of the box, decrease the reproducibility of experiments, handicap the ability to extend or adapt existing methods and increase the time required to understand the codebase. TIAToolbox is a suite of unit-tested image analysis and machine learning (ML) tools developed for the CPath community, making it possible for a variety of users to construct and reproduce CPath analytical pipelines with cutting-edge methods.

Our main objective is to provide an open-source library to the CPath community, which is simplified, streamlined, reproducible, easy to use, unit-tested and allows researchers to build their analytical pipelines on state-of-the-art methods. To achieve this, we provide a simple to use Application Programming Interface (API) which abstracts unnecessary complexity from the user where possible. This means that the API users can write code with a focus on the task at hand instead of being distracted by unnecessary details or peripheral tasks, such as managing multiple processes or needing to know the details of different WSI formats. The WSI reading capability of the toolbox is a good example of such abstraction that simplifies WSI reading. It hides unnecessary details of various file formats while keeping intact important format-related metadata required for ML tasks. For reproducibility of algorithms, we provide pretrained published benchmark algorithms which can be run using only a few lines of code. This can help researchers to build on state-of-the-art methods and greatly simplifies the reproduction of previous results. Weights for these pretrained models can be automatically downloaded at runtime or can be provided by the user, making it easier to test alternate models using the same pipeline. We posit that TIAToolbox will help establish objective and measurable standards of progress in the development of CPath algorithms.

One of our main guiding principles is to make CPath accessible to researchers without expertise in Deep Learning for CPath-specific tasks. We provide example notebooks (https://github.com/TissueImageAnalytics/tiatoolbox/tree/publication/examples) for this purpose. These notebooks can be run in a web browser on local machines or free-to-use platforms such as Google CoLab and Kaggle. The online platforms require no local installation and are well suited to non-technical users. The notebooks additionally serve as a manual by example for the use of the TIAToolbox. Our toolbox is supported by extensive online documentation (https://tia-toolbox.readthedocs.io/en/publication), including examples, for each module in TIAToolbox. In addition, we provide a command-line interface that enables experienced programmers to use the components of the package in Bash scripts and to batch-process their images or WSIs on CPU/GPU clusters.

In this section, we provide a brief review of existing tools for reading whole slide images (WSIs), image annotations, and image analysis. Image reading libraries, such as OpenSlide[5] and BioFormats[6], allow reading of WSI image formats. However, OpenSlide does not support several image formats. For example, it is unable to read JPEG-2000 JP2 images (although it can read JPEG-2000 J2K TIFF tiles) generated by legacy GE Omnyx scanners and images in OME-TIFF format (https://docs.openmicroscopy.org/ome-model/5.6.3/ome-tiff/), a commonly used open and well-documented file format. BioFormats supports reading of many WSI image formats. However, it is a Java library making it potentially difficult to integrate with Python-based workflows. The Java Python interface of BioFormats allows one to bridge this gap. However, it can be slow, complicated to set up and requires a variable set of parameters for different WSI formats—not ideal for a newcomer. Additionally, when reading JP2 images BioFormats relies on an outdated and unmaintained implementation from the Java Advanced Imaging (JAI) library for which support and documentation from Oracle has been discontinued. QuPath[7] provides a graphical user interface and the ability to read a variety of formats. However, because of its dependence on Java, its integration with a custom Python ML pipeline may require additional steps.

Although it is possible to use separate libraries for various formats, different interfaces and resulting data types can make writing code to handle multiple formats complex and error-prone; especially when trying to replicate existing algorithms. This causes a significant loss of researchers' time in handling technical issues instead of evaluating and developing new pipelines. There are other considerations, such as handling metadata from various formats, re-sampling of images, integration with image processing tools and optimizing data loading from machine learning libraries.

QuPath includes some classical image processing algorithms and also integrates with some DL models as plugins. For example, it includes a semantic pixel segmentation method which utilizes a user configurable set of simple image features (e.g., color channel intensity, gradient magnitude, Laplacian of Gaussian, etc.) which are fed to specified classifiers such as a random forest, k nearest neighbors (KNN), or artificial neural network (ANN). Pre-trained DL models, for example StarDist[8], are not included directly with QuPath but may be downloaded by a user and enabled as a plugin.

DL models typically produce results of higher quality than classical image processing methods, due to their ability to automatically extract representative image features. Therefore, we focus on including pre-trained cutting-edge pre-trained models in TIAToolbox which have been trained on images sampled across many slides using large public data sets, making them easily usable without any further user configuration or labelling.

Other Python software packages, such as PathML[9], offer some trained deep learning models. However, the selection is often limited, currently only one model (HoVer-Net) in the case of Dana-Farber-AIOS PathML, with a U-Net[10] implementation in progress. There is also no clearly documented way to integrate additional models or custom user models with PathML.

It is common for histology image analysis packages (such as HEAL[11], HistoCartography[12], and CLAM[13]) to focus on a particular method, model, or approach. In contrast, TIAToolbox can integrate with standard PyTorch modules (including many third-party PyTorch-based modules) and does not require the use of custom TIAToolbox layers or modules within the model architecture definition. It allows batch processing of several hundreds or thousands of WSIs and employs a modular structure, allowing for a wide variety of techniques to be integrated with the toolbox and for its modules to be used as components in new analytical pipelines.

TIAToolbox addresses the aforementioned issues and provides a broad feature set, shown in comparison with other histology focused software packages. The main contributions of TIAToolbox are as follows: development of histopathology image analysis pipelines, support for a wide range of WSI formats, a unified framework,

efficient image reads, tile generation (Zoomify), modularity, high unit-test coverage (>99%), reproducibility of state-of-the-art methods, cross-platform compatibility (Windows, Linux and macOS), ease of use, a command line interface (CLI) and a pure Python/ cPython source.

Our toolbox provides the most extensive integrated solution to a variety of important histopathology image analysis tasks ranging from multi-format image reading, patch and tile extraction, stain normalization, instance segmentation, patch classification and extraction of deep features for the development of WSI-level weakly supervised prediction models through weakly-supervised and graph neural network techniques as well as visualization of their results.

It has functionality to read common WSI image formats including OpenSlide compatible WSI formats (including Aperio SVS, Leica SCN), OME-TIFF (OMERO) and JP2 (Omnyx) in addition to visual fields (JPEG, PNG) using a single Python API in a unified framework. Furthermore, it also allows the addition of other existing and newly emerging formats.

Random-access reading and re-scaling of these WSIs based on resolution metadata (e.g., microns per pixel) is done efficiently, making use of multiple stored resolutions. This allows efficient implementation of multiple instance learning (MIL) algorithms such as IDaRS[3] that require random sampling of tiles. Designing the toolbox to be composed of modular re-usable components encourages the development of new analytical pipelines. We integrate and verify published models using these modules in addition to providing pretrained weights to enable reproduction of results.

We use abstraction where possible to reduce complexity for new users and to enable users with little programming experience to perform common tasks (such as shown in Supplementary Note 1) without having to worry about awkward edge cases. When implementing tools or integrating existing tools, we test for compatibility across Windows, Linux and macOS. In addition, we also provide many web-based example notebooks to run the code.

Lastly, there is no need to bridge between languages, such as between Java and Python. Only Python code or cPython compatible C/C++ extensions are used. Language bridges can be problematic to set up and often have performance issues. Therefore, we have avoided requiring one for the toolbox to function.

In summary, we present an open-source unit-tested, unified cross-platform software library with comprehensive tools for WSI reading, patch extraction, pre-processing, model inference, post-processing and visualization. We provide a platform for reproducible computational pathology using classical machine learning and advanced deep learning for end-to-end tissue image analysis.

## Methods

**Reading WSI data**. TIAToolbox provides a common interface for random-access reads of image regions from disk using an API defined in an abstract base class. Readers providing support for specific formats are implemented by sub-classing the base reader. We currently support reading a variety of tagged image file format (TIFF) based WSI images (including SVS, SCN, NDPI, MRXS and generic tiled TIFFs) using an OpenSlide[5] backend, OME-TIFF files using a tifffile (https://www.lfd.uci.edu/~gohlke/) backend and reading from JPEG 2000 based slide formats (such as JP2 files generated by GE Omnyx scanners) using the Glymur (https:// github.com/quintusdias/glymur) and OpenJPEG (https://www. openjpeg.org) as a backend. We also provide preliminary support for the rapidly evolving Zarr format (https://zarr.readthedocs. io/en/stable). Lastly, we include support for reading WSI DICOM images (via wsidicom) with JPEG and JPEG2000 compressed tiles. Furthermore, we include experimental support for a developing next generation file format (NGFF version 0.4) based on Zarr[14].

The reader class implements read functions based on physical resolution units, such as microns-per-pixel (MPP) or apparent magnification. This is useful to reproduce results of published algorithms which might have been trained at a specific magnification or MPP. For example, a read can be performed with the resulting image scaled to 0.5 MPP or an apparent magnification of 20×. For efficient image reads, we use pre-computed lower resolution copies when reading to avoid costly and unnecessary re-sampling of large image regions when re-scaling to the user requested resolution and units. This is done using metadata specifying the physical resolution of the WSI and down-sampled copies of the image embedded in the WSI file. The standard image pyramid is illustrated in Fig. 1, which depicts multiple copies of an image stacked on top of each other in decreasing resolution.

When reading a region from the WSI, we define two modes of operation: the *read_bounds* mode that allows reading with a fixed field of view as resolution varies and the *read_rect* mode with a fixed output size as resolution varies. The *read_rect* method accepts a location and output size as arguments. This method is useful for situations where the output must remain the same size, for example while extracting patches. As illustrated in Fig. 1, this results in a changing field of view as resolution varies. Conversely, *read_bounds* ensures a fixed field of view at all resolutions but may result in a different output image size. This is useful if there is some tissue feature which must be isolated in the view. To the best of our knowledge, no other tool provides equal flexibility in manipulating WSI pixel data.

Our advanced WSI reading tool easily fits within various CPath pipelines due to the wide range of image formats that it supports. This is demonstrated in the patch aggregator and graph aggregator pipelines as presented earlier, where the same reading functionality is incorporated. To help researchers easily use our toolbox for WSI reading, we provide a specific notebook (see Example Notebook 01) with multiple examples.

**Virtual whole slide image pyramid**. A virtual WSI reader class enables reading image data from single resolution visual fields, such as JPEG or PNG files, using the same interface as defined for reading WSIs. This facilitates the creation of a virtual image pyramid similar to the WSI pyramid in Fig. 1. An effective use case for this is when reading from an image derived from a WSI, such as a tissue mask or patch classification output map. These images are typically at a much lower resolution than the full-size WSI. A virtual image pyramid can have pyramid levels specified for which there is no stored re-sampled image, or which have larger dimensions than the image data itself. However, when read using the WSIReader interface, the virtual WSI will behave as if those resolution levels do exist simply by interpolating the available image data. As a result of this behavior, the original tissue WSI and a derived image can then be read synchronously, using the same coordinates and resolution arguments as shown in Supplementary Note 2, simply by copying the metadata about available resolutions and the physical scale (MPP) of the baseline resolution. This relieves the user of having to perform cumbersome and error-prone conversions between different coordinate systems.

**Metadata**. Metadata format varies greatly between file formats. We cater for this when initializing the reader object by creating a metadata object and thus providing a unified object when accessing image file related metadata. Since this is implemented as a Python class, static analysis tools common in many integrated development environments can parse it and offer helpful auto-completion suggestions, making it easier for researchers to write and implement their pipelines. The original underlying metadata

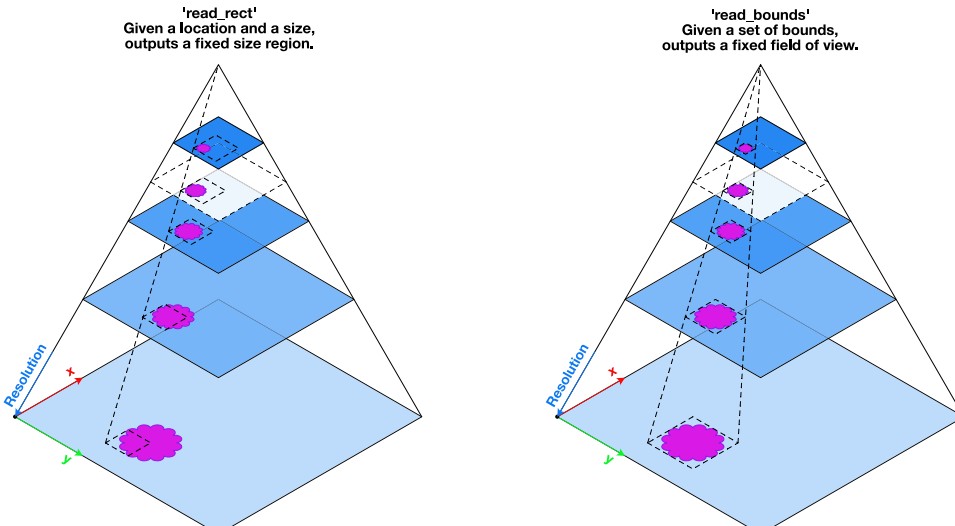

**Fig. 1 Illustration of two modes of random-access read from a multi-resolution (pyramidal) WSI.** Different resolutions, stored in the WSI, are shown as blue planes stacked on top of each other. A lower resolution is a stored down-sampled copy of the highest resolution (baseline). Here both read modes, read_rect and read_bounds, illustrate reading a region of interest containing some tissue (magenta shape) at a desired resolution. Reading of a region which is not at a pre-computed and stored resolution within the WSI (transparent white plane with a dashed outline) results in a read via a down-sample interpolation from a level with higher resolution.

is stored so that it remains accessible if required. Additionally, important metadata such as MPP may be specified if it is not found within the file. This is commonly useful when reading large visual fields or creating virtual WSIs which have a known magnification or MPP but are missing embedded metadata.

**Tissue masking**. Most WSIs contain a large amount of background area (e.g., glass, slide vendor name etc.) which is of no biological significance and can be ignored to speed up downstream processing and analysis. To identify these areas a tissue mask is commonly generated. In fact, tissue mask generation is common practice and is used in most CPath applications, including those presented in the Results section. We include some basic methods for creating such tissue masks based on Otsu thresholding[15], which separates pixels into foreground and background by minimizing the intra-class intensity variance. We show how one can combine Otsu's method with some basic morphological operations to remove small holes and regions. These masking classes can easily be extended to more advanced methods by creating a subclass of the abstract base class. A convenient function is provided to quickly generate a virtual WSI of a mask from a tissue WSI at a desired resolution. A notebook on tissue mask generation can be found within the TIAToolbox repository (see Example Notebook 03). We also provide a DL-based method for tissue masking, which is described in more detail later in the Semantic Segmentation section.

**Patch extraction**. It is common to apply DL models using small images[1,16] due to GPU memory constraints and required model complexity. As such, it is also common to divide a large WSI into small patches for training and inference with a model. This could be done simply by iterating over the WSI dimensions with a stride equal to the desired output patch size required by the model. However, there are several additional things to consider. Firstly, since pathology images are calibrated and have a known scale, patches may be extracted at a specific resolution (for example 0.5 microns-per-pixel). Our patch extractor rescales to the desired output resolution. Additionally, it can handle edge cases, such as whether to include patches which would partially extend beyond the edge of the WSI. Our patch extraction module can flexibly

handle such edge cases by either discarding these patches or padding to maintain a homogeneous output size. Also, an overlap can be specified so that each extracted patch partially overlaps its neighbors. The patch extractor, implemented as an iterator, can extract patches as needed which avoids filling available memory with patches until they are needed resulting in increased memory efficiency. In addition to grid-based patch extraction, patches may be extracted around each point in a set of coordinates. This is particularly useful for extracting patches centered on known cell nucleus locations or randomly distributed patches across the WSI. The *PatchExtractor* also supports functionality to filter out non-tissue regions while generating patches. To highlight the effectiveness of our efficient patch extraction tool, we provide an easy-to-follow interactive notebook with multiple examples (see Example Notebook 04).

**Stain normalization and augmentation**. It is well known that digital pathology images vary in their color appearance due to factors such as differences in scanner manufacture and variation in tissue preparation. For example, thicker specimens tend to stain the tissue darker. Differences in temperature, stain concentration, duration of staining and scanner type and settings can also lead to stain variation. This may harm the performance of automated methods, unless dealt with appropriately.

It is possible to perform simple color normalization using first-order statistical measurements but doing so may not correctly model the variation in stain appearance. A commonly used pathology specific pre-processing step is to perform separation of histological stains into separate optical density (OD) channels from the original red, green, blue (RGB) sensor data and optionally apply normalization across the OD channels. TIAToolbox includes several commonly used methods for normalization, including Reinhard[17], Macenko[18] and a modified Vahadane[19]. The toolbox implementation is adapted from the StainTools[20] Python package created by Byfield. Our implementation of Vahadane's method exchanges the SPArse Modelling Software (SPAMS)[21] dictionary learning step with an equivalent implementation in scikit-learn[22] and SPAMS LARS-LASSO regression with ordinary least squares (OLS) regression. We do this to maintain cross-platform compatibility and for speed of execution.

Other implementations of LARS-LASSO, for example in scikit-learn, performed orders of magnitude more slowly. We demonstrate how a user can use stain normalization in their pipelines by providing a descriptive Example Notebook (02).

Instead of normalizing image data, another method used in computational pathology is stain augmentation. This is particularly useful when training DL models to increase a model's robustness to stain variation. In TIAToolbox, we leverage stain extraction methods described above to randomly perturb the Hematoxylin and Eosin stain contents of each image used for training purposes. We also ensure integration of our stain augmentation functionality into commonly used augmentation packages, such as albumentations[23].

**Models**. Each CPath pipeline usually contains numerous steps and requires special consideration so that large-scale WSIs can be effectively dealt with. In fact, recent state-of-the-art models in computer vision for tasks such as segmentation[16] and classification[24,25] cannot be directly used when working with multi-gigapixel inputs due to memory limitations. This is due to the lack of available tools that can handle WSIs effectively in machine learning pipelines because of their high dimensionality. As the WSIs commonly get divided into smaller independent image patches, each processed by a machine learning model before merging the patch-level results, it is common practice to build custom tools from the bottom-up (i.e., starting from patches) to tackle such challenges.

Despite an increase in the number of models provided within CPath, model weights are not always available. Even when weights are provided, downloading and management can become challenging when working with multiple code repositories. Current DL libraries[26,27] enable seamless downloading of models, along with their parameters, yet these models have not been trained on problems within CPath. Even if these models were trained on task-specific data, additional work would still be required for use with WSIs.

To help overcome the above shortcomings, we provide an easy-to-use API where researchers can use, adapt and create models for CPath. TIAToolbox enables researchers with different levels of experience to easily integrate advanced CPath algorithms into their research projects. Once again, this avoids having to reinvent the wheel. We aim to achieve these goals by: Introducing a common API to assemble predictions for common CPath tasks, such as: instance segmentation, semantic segmentation and classification; integrating several well-established models (pretrained weights and model code) for the above tasks; utilizing a common data loader to seamlessly load WSIs within each model irrespective of the task at hand.

**API for models**. To enable integration of multiple models within the toolbox, we implement a common API, comprised of three components: a *Dataset Loader*, *Network Architecture* and *Engine*. The *Dataset Loader* defines how the data is sampled and converted into batches. The *Network Architecture* contains the model architecture, defines how to process an input batch and specifies how to post-process the results. An *Engine* defines how the *Network Architecture* and *Dataset Loader* interact, runs inference and assembles the output into a WSI-level prediction.

In the above three components, the *Dataset Loader* and *Engine* are designed in such a way that they should not need to be modified unless performing a task not supported by TIAToolbox. In our initial release, supported tasks include patch classification (*PatchPredictor*), semantic segmentation (*SemanticSegmentor*) and nucleus instance segmentation and classification (*NucleusInstanceSegmentor*). As described above, the *Network Architecture* defines the interaction of various network layers and determines how to transform the output into the final prediction via post-processing. We typically include the post-processing within the

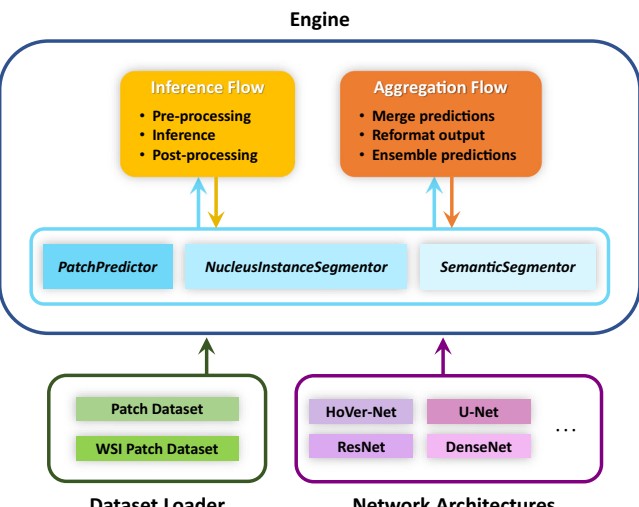

**Fig. 2 Diagram of the model(s) framework in the toolbox.** The framework comprises three main components: dataset loader, network architectures and engine.

network definition because this can often be model-specific. For example, nuclear instance segmentation models may produce different outputs and therefore need to be processed according to the type of output generated. We demonstrate how the *Dataset Loader*, *Network Architecture* and *Engine* interact in Fig. 2. Here, we observe that the *Dataset Loader* and *Network Architecture* are provided to the model *Engine*, where the data is then processed in the backend by the inference and aggregation flows.

In our toolbox, we support a handful of different models, such as ResNet[24] and DenseNet[25] for patch classification, U-Net[10] for semantic segmentation and HoVer-Net[1] for nuclear instance segmentation and classification. We also provide an extension of HoVer-Net that performs segmentation of additional regions using a single network[28]. We have designed the API in such a way that using a custom model in place of our supported models is straightforward. Therefore, researchers can focus solely on model development because the handling of WSI data is done behind-the-scenes by the *Dataset Loader* and *Engine*. With just a few lines of code, supported models can be used without modification. As part of our toolbox, we provide detailed examples that describe how to easily use both pre-defined and custom-built models for a given application.

Below we provide more information on the three main tasks initially supported in the toolbox: patch classification, semantic segmentation and nuclear instance segmentation and classification. The three tasks are similar in that they make a prediction for small image patches before aggregating the results. However, they differ in the type of output that is produced. For all these tasks, we provide detailed interactive example notebooks that clearly describe how to implement each of the models described in this paper. Sample outputs obtained using TIAToolbox for both semantic segmentation and nuclear instance segmentation & classification can be seen in Fig. 3.

**Patch classification**. Due to the sheer size of WSIs, DL methods in CPath often involve making a prediction based on smaller image patches. To assist with this, we provide a framework for patch-based classification, which can process image patches, larger image tiles or WSIs as input. Working with these different input types is streamlined in our toolbox and simply requires a user to define the input type as an argument in the code, as shown in Supplementary Note 3 and Supplementary Note 4. When the input is an image tile or WSI, the model will process each patch

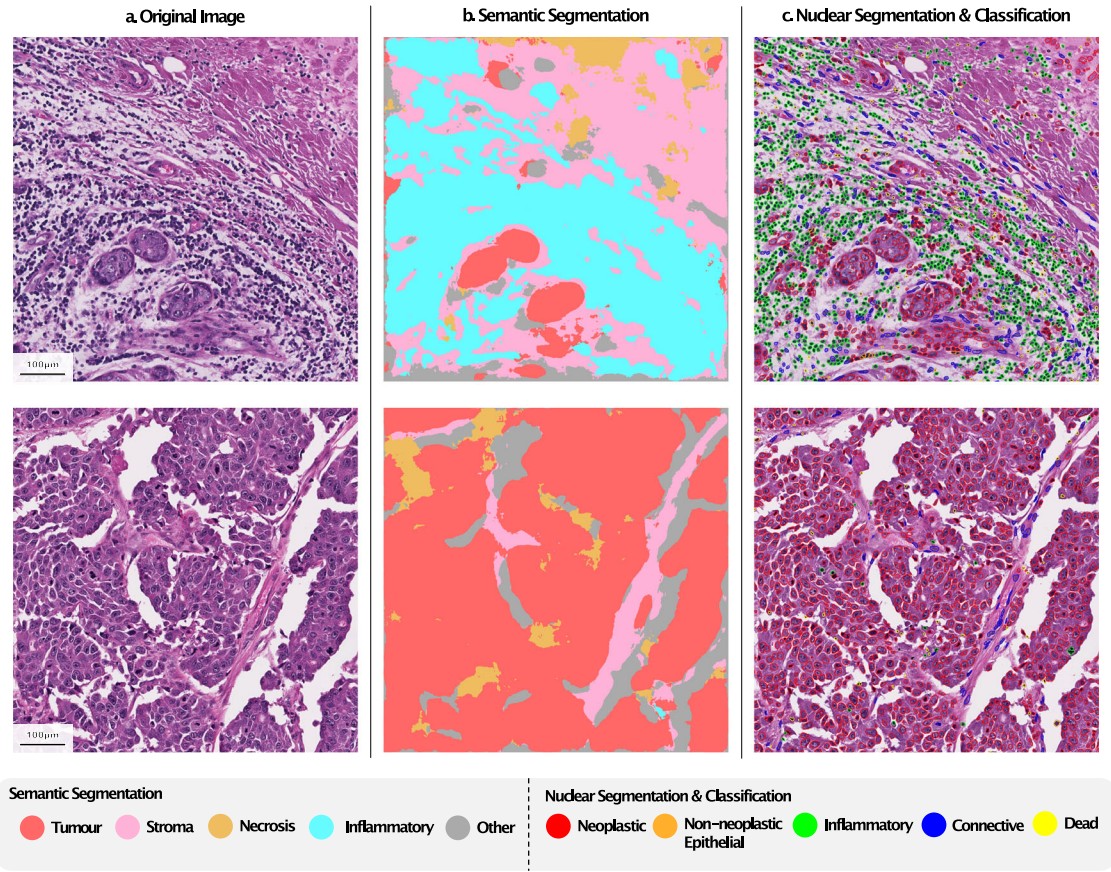

**Fig. 3 Illustration of simultaneous tissue Segmentation, nucleus classification. a** H&E stained input visual field. **b** Semantic segmentation output.
**c** Nucleus instance segmentation and classification output.

within the model consecutively and then aggregate the results to give a result for each patch within the input. The default post-processing scheme makes a patch-level prediction by selecting the class with the highest probability. The final output returns the path to a file that specifies the model predictions and the corresponding patch coordinates within each WSI image. When passing WSIs as input to our patch predictor, the toolbox internally uses the *PatchExtractor* class to obtain patches for the prediction model. Arguments for this extraction are passed through from the predictor initialization to the extractor.

We supply several pretrained models with TIAToolbox to allow users to process their data without the need to train their own models. We initially include models trained to predict different tissue types within colorectal cancer image patches, as introduced by Kather et al.[29], and models to classify breast tissue image patches as either normal or tumor[30,31]. When training models such as these we use publicly available train/test splits. However, it should be noted that for the Kather dataset we instead used a randomized 80/20 split on the non-stain-normalized data variant due to the availability of only stain-normalized test data. Non-normalized data was chosen to improve generality of the model.

When using models trained to predict the tissue type in colon tissue, the model will predict an input image patch to be one of the following nine classes: background, normal mucosa, tumor, inflammatory, debris, muscle, mucous, stroma or complex stroma. A full list of the available DL models for patch classification is given in Supplementary Table 1. For the breast tumor classification dataset, we used the PCAM training and validation splits. However, for the colorectal cancer dataset, we created our own data split to speed up the inference time. We show the validation results obtained after training each model on

the two patch classification datasets in Supplementary Table 2. We also highlight the ease of use of our patch predictor by integrating it within our example pipeline on the prediction of key mutations and molecular pathways.

**Semantic segmentation**. It is often desirable to localize regions within an image, rather than assigning a value to an entire input patch. This enables a more precise delineation of region boundaries and allows morphological features to be extracted from the tissue. Semantic segmentation localizes regions, without separating touching objects belonging to the same class. This may be sufficient when analyzing different tissue regions, such as tumor and stroma and the aim is not to extract subsequent features from individual objects, such as glands and nuclei. As in the case of the patch classification model, our semantic segmentation framework processes input patches separately, before merging the results. The difference here is that a prediction is made per pixel, rather than for the entire image patch. Despite this, the API remains similar between the patch prediction and semantic segmentation tools, as can be seen in Supplementary Note 5. The output of the model is a 2-dimensional map of the segmentation prediction, at a resolution specified by the user.

In the toolbox, we provide a U-Net based model with a ResNet50 backbone, trained on a multi-class breast cancer semantic segmentation (BCSS) dataset[32]. Here, the model will predict pixels to be one of: tumor, stroma, inflammatory, necrosis or other. In Supplementary Table 3, we report the Dice score for each class obtained by our model after training. We compare these scores to those obtained in the original paper and observe that overall, we achieve a better performance in terms of average dice score over all

classes. Note that despite the models not being identical, they both use a U-Net architecture with a ResNet50 backbone. In addition, we train the same model architecture for the task of tissue masking, which can enable a more precise result than conventional threshold-based methods (see Example Notebook 06).

**Nuclear instance segmentation and classification**. Identification and localization of different nuclei is a particularly important task in the field of CPath because it enables subsequent extraction of cell-based features that can be used in various downstream tasks, such as cancer grading[33] and biomarker discovery[34,35]. Identification and localization of different nuclei is a particularly important task in the field of CPath because it enables subsequent extraction of cell-based features that can be used in various downstream tasks, such as cancer grading[33] and biomarker discovery[34,35]. Here, it is necessary to separate clustered nuclei at the output of the model to ensure that features inferred from the model output correspond to individual nuclei. Classifying the types of nuclei can help profile the tumor microenvironment because it enables the quantification of different types of cells in various areas of the tissue. For this task, like other tasks defined in TIAToolbox, individual patches are processed before merging the results. However, a more complex post-processing step is needed to ensure individual nuclei are effectively separated and classified into distinct categories.

For this task, we provide a top-performing approach for nuclear instance segmentation and classification within TIAToolbox, developed by members of our research group. The model, named HoVer-Net, has been increasingly used in recent research projects[4,12] in CPath, due to its state-of-the-art performance across a range of different datasets. In the toolbox, we include nuclear instance segmentation models trained on the PanNuke[36,37], CoNSeP[1] and MoNuSAC[38] datasets—three widely used datasets for instance segmentation and classification of nuclei. For this, we use the original model weights and therefore, we encourage readers to refer to the original papers for details on performance. Further information on the predicted classes when using models trained on the aforementioned datasets is provided in Supplementary Tables 1-8. We demonstrate how easily our nuclear instance segmentation tool can be integrated into CPath pipelines by demonstrating how it can be seamlessly used during our graph aggregator example. Again, our nuclear instance segmentation and classification tool is simple to use and uses an API in line with other models in the toolbox. This can be seen in Supplementary Note 6.

In addition, we provide HoVer-Net+[28], which extends the original HoVer-Net model by adding a fourth decoder to perform the task of region-level semantic segmentation. In particular, the model that we provide in the toolbox has been trained on a private cohort of oral epithelial dysplasia WSIs to segment various nuclei (e.g., epithelial, inflammatory) and the different intra-epithelial layers. For further information on performance, we ask readers to refer to the original publication[28].

**Customizing models**. In the toolbox, we supply model architectures along with associated pretrained weights to enable models to be used out-of-the-box. However, it may be desirable to use one of our defined model architectures, but with different weights. For example, users may train a model on a different dataset, or a different training strategy may be used to obtain the weights. If a user would like to do this, the default pretrained weights may be overridden by simply adding the path to new weights as a class initialization argument. We show an example of how this can be done in Supplementary Note 7. Furthermore, TIAToolbox is flexible and is designed to allow users to add their own PyTorch compatible models for any of the tasks included within the toolbox. We provide sufficient examples in the form of

interactive notebooks (See Example Notebook 07) to detail the steps required for model customization.

**Deep feature extraction**. In many CPath pipelines, it is of interest to extract deep features from input images, which can be used for downstream tasks, such as clustering[39], patch classification[13] and graph-based learning[12,40]. Visualizing deep features can also help us to better understand which areas within an image the model may be focusing on, which can help further guide researchers with model development. Deep features are obtained by passing an image through a trained CNN and extracting the features immediately before the classification layers. A popular strategy is to utilize networks trained on the ImageNet dataset because they are optimized on millions of example images and thus are likely capable of extracting strong features. Therefore, we ensure that ImageNet-trained models can be integrated with TIAToolbox, enabling extraction of strong deep features for downstream tasks. In future, we plan to support extracting deep features using additional datasets and different optimization techniques such as self-supervised learning.

**Visualization**. We provide several convenient functions for visualizing the results of model predictions. These include merging of prediction outputs and overlaying predictions on the predictor input image (Example Notebook 05) and plotting a generated graph (see Supplementary Note 8). Our toolbox also implements generating multi-resolution tiles in a format commonly used by interactive web-based (slippery map) viewers such as OpenLayers (https://openlayers.org) where a tile server streams image regions on-demand to a web client, for display of very large images and geospatial data which can be panned and zoomed by a user. We additionally include a simple web application that can be viewed in a web browser. An example of this is shown in Supplementary Note 9. This can also be used in combination with the functionality of a virtual WSI to allow for ease of visualization, such as overlaying patch predictions on top of a WSI.

**Annotation storage**. It is common for CPath algorithms to output geometric annotations such as cell or gland boundaries along with some associated properties such as a class label or certainty metric. In this paper and in our toolbox, we refer to this combination of a geometric entity and its associated properties as an annotation. A geometric entity may be a point, polygon, sequence of line segments (line string) or a closed line string loop with no area (linear ring). Properties are defined to be a hierarchical JSON-like structure which may contain strings, integers, floats, dictionaries and lists.

Storage and retrieval of annotations are non-trivial due to the potentially enormous number of detected geometric entities, which may be several million in the case of nuclear boundaries, for just a single WSI. It is often infeasible to keep many annotations in the memory of a desktop workstation. Furthermore, searching for relevant annotations when performing downstream analysis may be slow if a naïve methodical scanning method is used. To address these issues and complement the output of nuclear segmentation models, we implement an annotation storage class that can efficiently handle a large number of geometric entities and their associated properties. We implement a base class which defines an interface extending the standard Python *MutableMapping*. This enables users to interact with our storage classes using regular Python syntax and idioms, much like working with a standard dictionary (hash table) object, thus avoiding the burden on a user to learn a new set of functions and interaction mechanics.

We provide two concrete implementations of this storage class. One that is backed by a simple in-memory dictionary (hash table) and is well suited for small annotation sets. The other uses an

SQLite database to store geometric entities as well-known binary (WKB) and properties as JSON. Example usage of the SQLite annotation store class is shown in Supplementary Note 10. The SQLite store implements several optimizations to make it suitable for large annotation sets. Primarily an R-Tree index is used which enables fast spatial queries using bounding boxes, providing a significant improvement over naively testing every annotation in the store for intersection on each query. We extend this simple bounding box query to full binary shape predicate testing via a registered custom function call-back to Python which acts as a secondary filtering stage after the initial bounding box query. This allows querying for only annotations which intersect with any arbitrary polygon. Furthermore, we utilize a restricted subset of the Python language to provide a simple domain-specific language (DSL) and thus enabling predicate statements to be supplied to queries and evaluated in an efficient manner for a specific backend where possible. For example, when querying annotation properties from an SQLite store, it is possible to check entries in the properties of an annotation as part of the SQL query itself. This can evaluate the query in the highly optimized SQLite query environment. It can also avoid the potentially costly decoding of the full properties on top of a roundtrip to the Python interpreter for evaluation. We also provide a fallback to a simple post-query filtering in Python should there be no optimization available. Lastly, this store enables compression (via the zlib library) of WKB geometry, considerably reducing the space required to store geometries but at the cost of additional encoding and decoding time. Point annotations are an exception to this as no WKB is stored for them. The required R-Tree index row for a point annotation contains all necessary information to recreate a point and therefore no additional storage is used.

Some convenience functions are provided for converting to and from various formats. A store may be created from or exported to several formats including a Pandas DataFrame, GeoJSON feature collection, Line-delimited JSON (ndjson), or Python dictionary. We expect our annotation store to contribute to the standardization of AI-generated annotations.

To demonstrate the performance benefits provided by our SQlite backed storage class, we provide a benchmarking notebook. This notebook performs several common tasks on a dataset of over 5 million generated cell boundary polygons.

**Ethical approvals for datasets**. We built our software on datasets which are previously published or publicly available. Therefore, no additional ethical approval was required for this paper.

**Reporting summary**. Further information on research design is available in the Nature Research Reporting Summary linked to this article.

## Results

In this section, we demonstrate the utility of TIAToolbox for two WSI-level prediction tasks, using recently proposed DL models, while demonstrating several of the other functionalities of the toolbox. First, we predict the status of molecular pathways and key mutations in colorectal cancer from Hematoxylin and Eosin-stained (H&E) histology images using a two-stage patch-level classification model. Next, we predict the HER2 and ER status from H&E histology images using SlideGraph+, a graph neural network-based model. We show that the implementation of both the pipelines has been simplified using a common interface provided by TIAToolbox as shown in Fig. 4. This reduces the effort needed by a new researcher seeking to extend these approaches.

**Patch aggregator: predicting the status of molecular pathways and mutations using patch-level predictions**. Assessment of the status of molecular pathways and key genetic mutations helps better understand the patient prognosis and can provide important cues for treatment planning. Typically, this assessment is done via genetic (e.g., polymerase chain reaction or PCR) or immunohistochemistry (IHC) testing. However, these tests may lead to time delays and additional costs because they are used as an extra step after initial analysis on routine H&E-stained slides. Recently, it has been shown that deep learning has the potential to predict the status of pathways and mutations directly from the H&E slides, potentially bypassing the need for additional tests[3,41].

Despite the obvious advantages of H&E based prediction using deep learning, some researchers may struggle to reproduce the mutation prediction pipeline, where slight changes in the code may lead to much different results. Furthermore, new researchers may be discouraged from implementing the method, due to the challenge of working with high dimensional histology data. Here, we show that TIAToolbox can be used to complete all necessary steps to predict the status of molecular pathways and key mutations in colorectal cancer and help simplify the overall analytical workflow. To achieve this, we follow the same approach used in the original paper by Bilal et al.[3] and use a two-stage pipeline. We first localize the tumor regions to identify the potentially diagnostic areas and then use the IDaRS model of Bilal et al.[3] to make a prediction for the entire whole-slide image. Using the toolbox, these two steps can be completed with reproducible results without the need for advanced programming experience. We display our entire simplified IDaRS integration into TIAToolbox in Supplementary Note 11. It is worth noting that both stages use the toolbox's *PatchPredictor*, as shown in Supplementary Note 4 and differ only in terms of the pretrained model, which is defined during class initialization.

Identifying the tumor regions as an initial step is important for various tasks, for instance enabling the downstream analysis to be focused on diagnostically relevant areas. This initial step may also be useful in other tasks, such as cancer staging[42] and cancer subtyping[43]. To help overcome challenges resulting from limited computer memory, it is common to divide multi-gigapixel WSIs into smaller image patches, which are processed independently before merging the results. Using this approach, we obtain a tumor detection map by determining whether each input patch within the tissue contains any tumor. We utilize a pretrained ResNet[24] within TIAToolbox's *PatchPredictor* model to efficiently deal with patch-level processing and aggregation.

After obtaining the tumor detection map, we follow a similar divide, process and merge approach to obtain the task-specific prediction map. Using TIAToolbox's patch prediction functionality, each tumor patch is seamlessly processed with a pretrained ResNet and the results are merged. This prediction map can help improve the interpretability of results made by IDaRS and identify areas contributing to the overall prediction. To obtain the final WSI prediction, patch results are aggregated to give a single score. IDaRS is a weakly-supervised approach, trained using multi-instance learning technique and therefore the slide-level score is obtained using a common pooling strategy, such as selecting the maximum or average probability over all tumor patches. In the toolbox, we provide models trained on the first fold used in the original paper by Bilal et al.[3] to predict the following: microsatellite instability, hypermutation density, chromosomal instability, CpG island methylator phenotype (CIMP)-high prediction, *BRAF* mutation and *TP53* mutation.

As a result of TIAToolbox taking care of complex WSI handling behind the scenes, this pipeline has been reproduced in Example Notebook (IDaRS), utilizing the same code fragment as in Supplementary Note 4 with the toolbox as the

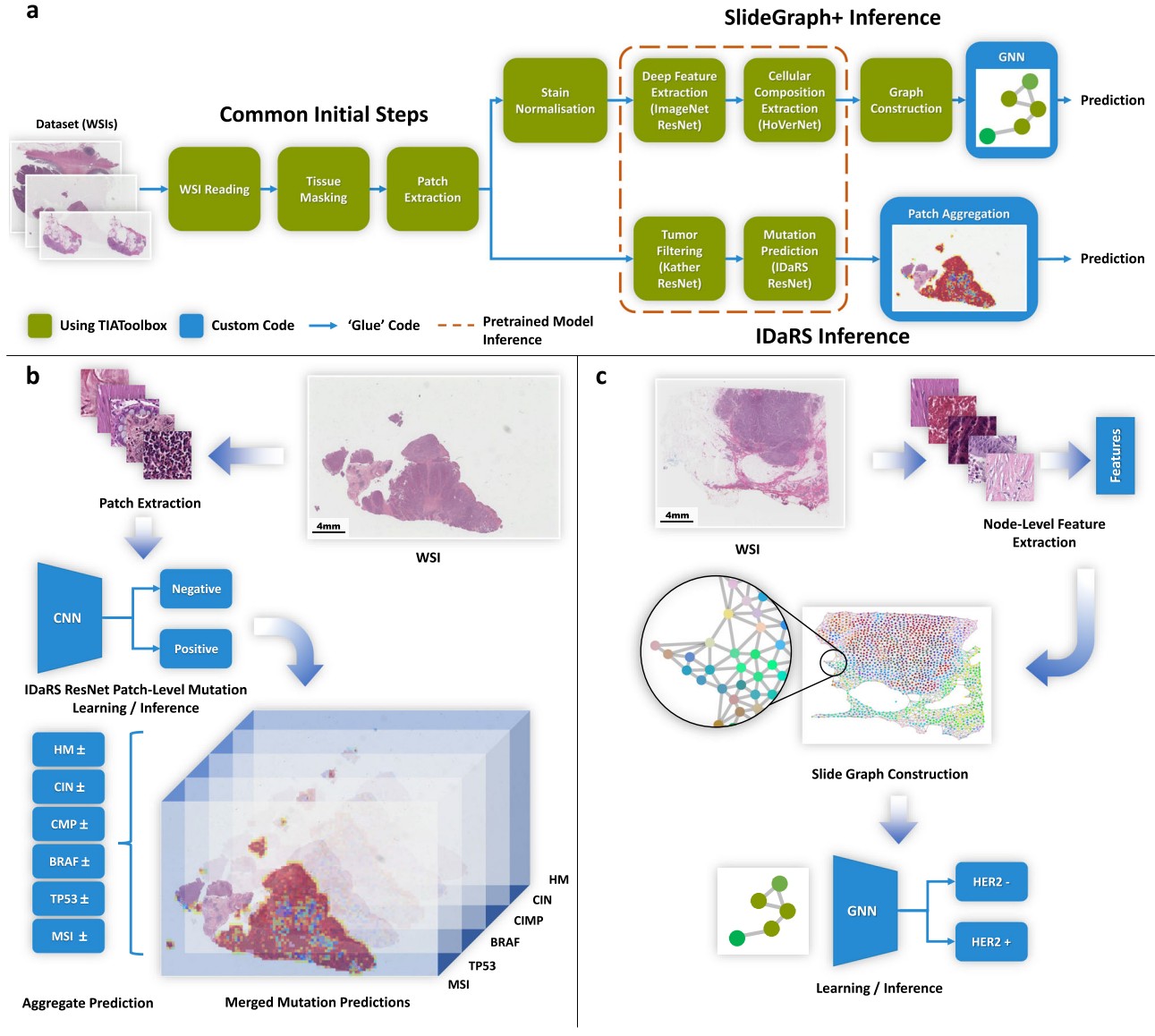

**Fig. 4 Example usage of TIAToolbox modules in AI pipelines.** The diagram shows the main steps of the SlideGraph and IDaRS pipelines and how modules in TIAToolbox have been used to replicate these pipelines. **a** Simplified block diagram of the main steps involved in each of the example pipelines. Several of the steps at the start of these pipelines are common between the two methods and are provided by TIAToolbox. Additionally, many of the steps where the pipelines diverge are also included in the toolbox. Custom code is only required for one or two steps in each pipeline in addition to gluing together each of the pipeline stages, or for some custom visualization. The same pretrained models can be used for inference in both IDaRS and SlideGraph+ pipelines. **b** The main steps of the IDaRS pipeline for an example WSI. For each input patch a mutation prediction (positive or negative) is made and the results merged. Each component of the output vector is represented as a plane in a stack. **c** An example WSI and the resulting graph from the SlideGraph+ pipeline. Nodes are colored in RGB space via a uniform manifold approximation and projection (UMAP) of the feature vectors assigned to the nodes.

backend uses significantly fewer lines of code than the original implementation. This highlights how functionalities in the proposed toolbox can be efficiently leveraged for WSI prediction tasks in CPath. These patch prediction models can use individual patches, larger image tiles or WSIs as input. For this example and to follow the approach used by Bilal et al.[3], we choose to focus on WSI-level inputs. To help reduce the inference time, the models that we include within the toolbox have been retrained without stain normalization, as opposed to the original IDaRS implementation. A full breakdown of performance obtained after retraining is provided in Supplementary Table 4. We observe that despite a slight reduction in performance which may be due to not using stain normalization, models provided with the toolbox can successfully predict molecular pathways and mutations.

**Graph aggregator: predicting HER2 status using SlideGraph+.** HER2 and ER status are key prognostic indicators for establishing an appropriate breast cancer treatment plan. As with other biomarkers, they are typically assessed with IHC staining. Instead, determining status via routine H&E slides can potentially reduce costs and time-to-treatment. We show the integration of SlideGraph+[40] pipeline using TIAToolbox for the prediction of HER2 status and ER status from H&E-stained histopathology images. SlideGraph+[40] is a message-passing graph neural network-based pipeline for WSI-level prediction that works by modelling each WSI as a graph with nodes corresponding to tissue regions and each node having a set of local features. Edges between nodes represent spatial organization within the tissue (see Fig. 4).

The SlideGraph+ pipeline consists of patch extraction from WSI(s), stain normalization, node-level feature extraction, graph

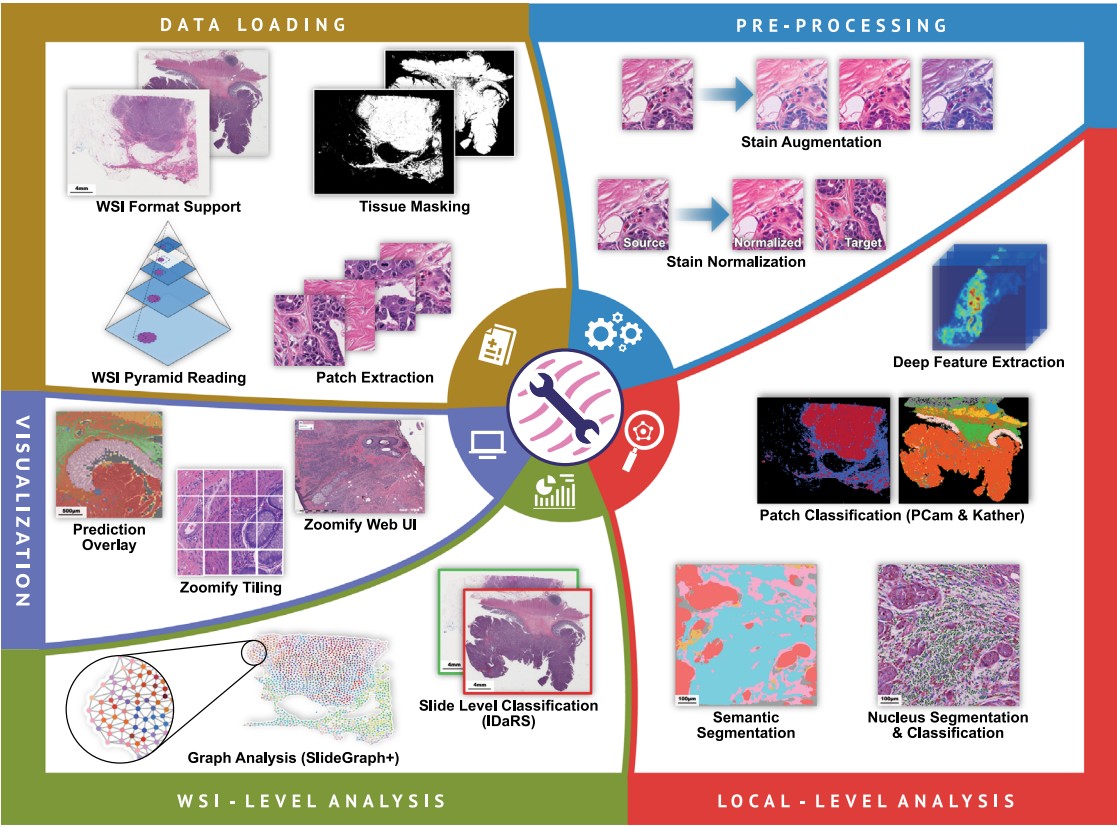

**Fig. 5 An illustration of the broad range of capabilities of TIAToolbox.** Capabilities of the toolbox are shown segmented into categories of data loading, pre-processing, local-level analysis, WSI-level analysis and visualization.

construction and prediction of the WSI label via a graph convolutional network (GCN). It is perhaps worth noting that this graph-based method is generic and can be applied to a wide range of WSI classification problems, as it is agnostic to both the problem at hand and the features utilized for prediction.

As shown in Figs. 4 and 5, the IDaRS and the SlideGraph+ pipelines have numerous modules in common. Many of the same modules used in the IDaRS pipeline can be reused without reimplementation of the whole pipeline. Using *PatchExtractor* and *StainNormalizer*, it is easy to extract patches from tissue regions of the WSI and apply stain normalization across patches in the same way that is done in the original SlideGraph+ implementation. For each of these patches, a set of features must then be extracted. Different types of features can be extracted here, such as deep features from a CNN pretrained on the ImageNet dataset, cellular morphological features (class, major axis diameter, eccentricity, etc.) derived from the HoVer-Net segmentation and classification output, or the output of a network trained to regress from an H&E patch to the corresponding DAB intensity between registered H&E and IHC slides, as demonstrated by Lu et al. In our example implementation, we use TIAToolbox's *DeepFeatureExtractor* to obtain features from an ImageNet-pretrained ResNet and cellular morphology features from HoVer-Net. Here, the state-of-the-art HoVer-Net model is provided as part of the toolbox's *NucleusInstanceSegmentor* engine, which can be used to subsequently obtain either deep or cellular composition features. The modularity of the toolbox and the flexibility of the SlideGraph+ method allows for fast and easy experimentation, without having to write a lot of code to reimplement many common steps like patch extraction, stain normalization and feature extraction.

TIAToolbox provides a hybrid clustering graph construction method, as used by Lu et al.[40], which requires only the location of

each patch within the WSI and a corresponding node-level feature vector. This method clusters patches based on a weighted combination of location and extracted features such that regions with similar features or locations in a WSI are grouped into the same cluster.

The extraction of features and construction of WSI graph representations by TIAToolbox can be easily integrated with code for training a GCN. The modular nature of TIAToolbox allows for easy integration into a Jupyter notebook as part of the toolbox examples to successfully reproduce the SlideGraph+ results obtained in the original Lu et al. paper[40] using ImageNet deep features and HoVer-Net derived cellular morphology features. TIAToolbox also enabled the ER status prediction using the SlideGraph+ methodology. For a full breakdown of these results and comparison with the original results, refer to Supplementary Table 5. Here the results that we report are obtained using five-fold cross validation.

**Discussion**

TIAToolbox aims to ease the handling of WSI data for analysis and visualization by providing an easy-to-use API that enables seamless reading, pre-processing and analysis of digital pathology slides. Therefore, we hope this will enable the users of TIA toolbox to access a wide and comprehensive set of tools, enabling them to focus more on model development.

Despite the rapid advancement of CPath, there has been no unified software library tailored towards the large-scale batch processing and analysis of pathology slides using state-of-the-art DL models. Previous packages have focused on a smaller subset of features, such as stain normalization or WSI reading. As can be seen in Table 1, TIAToolbox is an extensive library in terms of the

**Table 1 Comparison of features available in different histopathology image analysis focused software packages.**

| | TIA Toolbox | HistoCarto Graphy[12] | HEAL[11] | QuPath[7] | PathML[9] | CLAM[13] | Multi_Scale_Tools[51] | stainlib† | IBM CODAIT‡ | HistoQC[2] | Histomics[52] | Cytomine[53] |
|---|---|---|---|---|---|---|---|---|---|---|---|---|
| Platform Compatibility | Windows, Linux, Mac | ? | Docker (Linux) | Windows, Linux, Mac | ? | ? | ? | ? | ? | ? | Linux, Windows | Web/Cloud/Linux |
| Language | Python | Python | Python | Java (+Groovy) | Java +Python | Python | Python | Python | Python | Python | Python | Java/Python |
| Modular Usage | ✓ | ✓ | ✗ | ✓ | ✓ | ✓ | ✗ | ✓ | ✗ | ✗ | ✓ | ✗ |
| Unit Test Coverage | >99% | 88% | 0% | ? | 87% | 0% | 0% | 0% | 0% | ? | 72% | >70% |
| Read Visual Fields (PNG, JPEG) | ✓ | ✓ | ✓ | ✓ | ✓ | ✓ | ✓ | ✓ | ✓ | ✗ | ✓ | ✓ |
| Read TIFF (SVS, SCN, MRXS) | ✓ | ✗ | ✓ | ✓ | ✓ | ✓ | ✓ | ✗ | ✓ | ✓ | ✓ | ✓ |
| Read JP2 (Omnyx) | ✓ | ✗ | ✗ | ✗ | ✗ | ✗ | ✗ | ✗ | ✗ | ✗ | ✓ | ? |
| Read OME-TIFF | ✓ | ✗ | ✗ | ✓ | ✓ | ✗ | ✗ | ✗ | ✗ | ✗ | ✓ | ✓ |
| Read DICOM WSI | ✓ | ✗ | ✗ | ✓ | ✓ | ✗ | ✗ | ✗ | ✗ | ✗ | ✓ | ✓ |
| Read Zeiss CZI | ✗ | ✗ | ✗ | ✓ | ✓ | ✗ | ✗ | ✗ | ✗ | ✗ | ✓ | ? |
| Read Olympus VSI | ✗ | ✗ | ✗ | ✓ | ✓ | ✓ | ✗ | ✗ | ✗ | ✗ | ✓ | ? |
| Fluorescence Imaging | ! | ✗ | ✗ | ✓ | ✓ | ✗ | ✗ | ✗ | ✗ | ✗ | ✓ | ? |
| Physical Unit Based Reads | ✓ | ✗ | ✗ | ✗ | ✗ | ✗ | ✗ | ✗ | ✗ | ✗ | ✗ | ✓ |
| Tissue Masking/Detection | ✓ | ✓ | ✗ | ✓ | ✓ | ✓ | ✗ | ✗ | ✓ | ✓ | ✓ | ✓ |
| Patch Extraction | ✓ | ✗ | ✓ | ✓ | ✗ | ✓ | ✓ | ✗ | ✓ | ✗ | ✓ | ✓ |
| Stain Separation | ✓ | ✓ | ✗ | ✓ | ✓ | ✗ | ✗ | ✓ | ✗ | ✗ | ✓ | ? |
| Stain Normalization | ✓ | ✓ | ✓ | ✗ | ✓ | ✗ | ✗ | ✓ | ✗ | ✗ | ✓ | ✗ |
| Stain Augmentation | ✓ | ✗ | ✗ | ✗ | ✗ | ✗ | ✗ | ✓ | ✗ | ✗ | ✗ | ✗ |
| Patch Classification | ✓ | ✗ | ✗ | ✓ | ✗ | ✗ | ✓ | ✗ | ! | ✗ | ✗ | ! |
| Tissue Semantic Segmentation | ✓ | ✗ | ✗ | ! | ✓ | ✗ | ! | ✗ | ✗ | ✗ | ✗ | ! |
| Nucleus Segmentation | ✓ | ✓ | ✗ | ✓ | ✓ | ✗ | ✗ | ✗ | ✗ | ✗ | ✓ | ! |
| Nucleus Detection | ✓ | ✗ | ✗ | ✓ | ✓ | ✗ | ✗ | ✗ | ✗ | ✗ | ✓ | ! |
| Nucleus Classification | ✓ | ✗ | ✗ | ✓ | ✓ | ✗ | ✗ | ✗ | ✗ | ✗ | ✗ | ! |
| Deep Feature Extraction | ✓ | ✓ | ✗ | ✗ | ✗ | ✓ | ✗ | ✗ | ✗ | ✗ | ✓ | ! |
| Graph Learning | ✓ | ✓ | ✗ | ✗ | ✗ | ✗ | ✗ | ✗ | ✗ | ✗ | ✗ | ✗ |
| Whole Slide Classification | ✓ | ✓ | ✓ | ✗ | ✗ | ✓ | ✗ | ✗ | ✗ | ✗ | ✗ | ✓ |
| Graphical User Interface (GUI) | !⁋ | ✗ | ✗ | ✓ | ✗ | ✗ | ✗ | ✗ | ✗ | ! | ✓ | ✓ |

An exclamation mark (!) indicates a feature that may be partially implemented or is possible with the software, but either requires training or is not directly integrated with the software package. A question mark (?) indicates that there may be an appropriate metric, but no reported value could be found or support for this feature is unclear. †stainlib source at https://github.com/sebastianffx/stainlib, ‡IBM CODAIT deep-histopath source at https://github.com/CODAIT/deep-histopath. ⁋TIAToolbox includes a basic web-based UI for displaying WSIs on top of each other with adjustable opacity, but not a full-featured GUI.

number of features that it supports and therefore, we hope that CPath users will choose to use it for various applications in CPath.

We demonstrate the utility of TIAToolbox by using its core constituents to reproduce the results of two state-of-the-art AI pipelines in CPath. First, we predict the status of molecular pathways and key mutations in colorectal cancer and then we predict with SlideGraph+ the HER2 status from H&E-stained histology images. These pipelines have been implemented in the form of interactive notebooks, which can be opened and evaluated on cloud platforms such as Google Colab and Kaggle. This highlights how the toolbox can be used to substantially simplify previously complex approaches in CPath. We hope that the examples provided will motivate others to integrate the tools provided by TIAToolbox into their pipelines and help accelerate the development of new methods in CPath. The design of the toolbox ensures that the API remains consistent and easy-to-use when introducing additional models and tools. The two presented pipelines are algorithmically different, but due to the modular nature of the toolbox, code segments could easily be shared between each method, as highlighted in Fig. 4 where we observe that the first few steps are common due to re-use of TIAToolbox modules. Both pipelines can also share the same model inference code as highlighted in the figure. For example, all pipelines that use WSIs as input will use our advanced image reading functionality that supports a wide range of WSI formats, including JP2 and those supported by OpenSlide. Also, batch processing and patch aggregation are handled behind the scenes in both pipelines, without exposing unnecessary detail to the user.

We stress that TIAToolbox is not limited to the above two tasks and despite it being desirable to use our toolbox within all steps of a CPath pipeline, this is not a requirement. Due to its modular and extendable design, individual steps and various utility functions can be used in isolation for a broad range of applications in CPath. This helps in training new customizable algorithms on top of existing work. For example, dividing WSIs into patches before aggregating results is a widely used approach in CPath and this procedure is fully handled within the toolbox. Therefore, any pipeline that involves patch-level processing will benefit from the functionality that we provide. In fact, any patch prediction or segmentation model, based on PyTorch, can be integrated because our API is consistent irrespective of the model choice. The toolbox is not limited to the pretrained models that we provide. Any model trained outside our toolbox can be seamlessly integrated. We have demonstrated this flexibility with the help of a notebook (see Example Notebook 07) that uses natural images from the ImageNet data set. This enables one to utilize our toolbox for a large array of tasks in CPath, such as: cancer staging[42], cancer subtyping[44], survival analysis and the prediction of additional molecular pathways[44]. Additional tools can also be leveraged, such as efficient patch extraction, tissue mask generation, visualization and stain normalization, which can all be important steps in the automated analysis of WSIs.

TIAToolbox is available as a PyPi package (via 'pip install tiatoolbox'), conda-forge (via 'conda -c conda-forge install tiatoolbox') package, and as a Docker container via the GitHub container registry ('docker pull ghcr.io/tissueimageanalytics/tiatoolbox:latest').

TIAToolbox is an open-source project, to which additional pretrained models and features will continue to be added. In future, we will extend the currently available models by training on new datasets, increasing the number of applications of our toolbox. A logical extension would be to train and provide patch prediction models for colon cancer grading[45] and tumor detection in additional tissue types. We also aim to provide instance segmentation, detection and classification models for tissue structures such as glands, blood vessels and nerves, enabling the extraction of further interpretable morphological features for downstream analysis such as linking these features to survival or investigating spatial profile of the tumor microenvironment (TME). To enable a better understanding of how models are interpreting images, we aim to include tools that enable visualization of model activation maps on images. This can be done via techniques such as class activation maps (CAM)[46]. Currently, our SlideGraph+ pipeline utilizes functionalities from various parts of the codebase and integrates them into a notebook. In future, we plan to fully integrate a graph predictor engine within the toolbox, in addition to our existing patch predictor, semantic segmentor and nucleus instance segmentor engines. Going forward, the TIAToolbox could act as an enabler for commercial growth and encourage the use of CPath applications in a clinical setting. We anticipate and encourage users to contribute new features and integrate the provided tools into their own CPath pipelines to accelerate development of CPath as a field.

## Data availability

All datasets analysed during the production of TIAToolbox, except for one private oral dysplasia cohort dataset for HoVer-Net + , are publicly available. They can be accessed for research and non-commercial use at the following web addresses: The Cancer Genome Atlas (TCGA)[47] available at https://www.cancer.gov/tcga, PanNuke[36,37] available at https://warwick.ac.uk/fac/cross_fac/tia/data/pannuke, PatchCamelyon (PCam)[30] available at https://github.com/basveeling/pcam, Kather 100k[29,48] available at https://zenodo.org/record/1214456, Kumar (MoNuSeg Subset)[49] available at https://monuseg.grand-challenge.org/, MoNuSAC[38] available at https://monusac-2020.grand-challenge.org/ and CoNSeP[1] available at https://warwick.ac.uk/fac/cross_fac/tia/data/hovernet/. The private oral dysplasia cohort dataset is not available because we do not currently have ethical approval to share this dataset publicly but the trained model is already published with ethical approval details listed in the original publication[28].

## Code availability

All source code for TIAToolbox is available on GitHub (https://github.com/TissueImageAnalytics/tiatoolbox/tree/publication) and Zenodo[50] (https://doi.org/10.5281/zenodo.6808365) under the BSD 3-clause license. Model weights downloaded at runtime are publicly hosted and maintained on TIA Centre servers under a creative commons non-commercial use license (CC-BY-NC 4.0). All parts of the toolbox, including model weights, may be freely used for research and non-commercial purposes. Model weights can be made available for commercial use on request depending on ethical approvals from the data source.

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

## Acknowledgements

S.G., M.J., G.H., M.B., W.L., S.R., F.M., and N.R. would like to acknowledge support from the PathLAKE digital pathology consortium which is funded by the Data to Early Diagnosis and Precision Medicine strand of the government's Industrial Strategy Challenge Fund, managed and delivered by UK Research and Innovation (UKRI). Q.D.V. was supported by the Royal Marsden Charity, S.D. and R.M.S.B. by the Chancellor's Scholarship at Warwick and AS by Cancer Research UK. We would like to give thanks to Peter Byfield for the StainTools[20] code (https://github.com/Peter554/StainTools) used for the stain extraction and normalization modules of the toolbox. In addition, the authors would like to acknowledge the following datasets used for training models supplied in TIAToolbox: Kather100K, PCam, CoNSeP, PanNuke and MoNuSAC.

## Author contributions

J.P. and S.G. contributed equally to this work. J.P., S.G., Q.D.V., M.J., A.S., and S.E.A.R. contributed to the development of TIAToolbox. J.P., S.G., Q.D.V., M.J., and D.E. contributed to documentation and wrote the example notebooks. S.D., G.H., and R.M.S.B. provided support for the development of code and helped with bug fixing and robust testing of the toolbox. M.B. and W.L. contributed towards the IDaRS and SlideGraph+ pipelines. J.P., S.G., and S.E.A.R. wrote the manuscript with contributions from all the authors. S.E.A.R. led and jointly supervised the project with input from FM and NMR.

## Competing interests

The authors declare the following competing interests: S.G. and N.R. are co-founders of Histofy Ltd. All other authors have no competing interests to declare.

**Additional information**

