## [Peer Review File · Communications Medicine]

Reviewers' comments:

Reviewer #1 (Remarks to the Author):

This manuscript introduces TIAToolbox, a new Python package for computational pathology.

The package looks to be very professional, with excellent documentation and test coverage. It contains general-purpose solutions to many of the often fiddly and frustrating aspects of developing analysis methods for whole slide images, such as extracting arbitrary or fixed-sized regions at specific resolutions and storing regions of interest. It also provides numerous core processing methods (e.g. for stain normalization), and provides a collection of pre-trained models. It brings together numerous disparate libraries (e.g. OpenSlide, scikit-image, scikit-learn, tiffile, PyTorch) to provide a unified API designed for researchers and data scientists working on computational pathology applications, particularly for H&E images. The manuscript describes two state-of-the-art methods reimplemented using TIAToolbox.

There are a few areas in which I believe the presentation could be improved:

1. The target audience and limitations of TIAToolbox are not clearly articulated, to the extent of giving a misleading impression of what it does. For example, the abstract describes TIAToolbox as 'a Python toolbox designed to make CPath more accessible to new and advanced CPath scientists and pathologists alike'. The inclusion of 'and pathologists alike' makes this a strong claim, that I do not think is fully justified. Making use of TIAToolbox relies heavily on strong coding and data science skills; it is not a complete software application that provides a user-friendly interface suitable for routine use by most pathologists or by non-programmers from other disciplines.

The later statement that 'users can use our toolbox as a "one-stop shop" for CPath' seems excessive for the same reasons.

2. Related to the above, Table 1. is strongly biased in favor of showing TIAToolbox as the most comprehensive CPath solution, listing only features available in the toolbox. For example, file formats such as CZI and DICOM that are not supported by TIAToolbox are omitted, along with other potentially key features (e.g. fluorescence/multiplexed image support, interactive GUI, whole slide viewer, cross-platform installer). Given the lack of clarity regarding TIAToolbox's target audience and limitations, this selectivity could be considered to misrepresent TIAToolbox in relation to other software. I believe it would be preferable to position the toolbox more accurately within the community of open-source tools - where no one project does everything, but nor does it necessarily have to - rather than as more comprehensive than all others.

3. The 'Results' section focusses on two WSI prediction tasks using DL models recently proposed by members of the authorship team. This lacks clear metrics indicating the benefits for TIAToolbox; there is a reference to using 'significantly fewer lines of code', is it possible to provide numbers for this? Or perhaps performance comparisons, at least on a given hardware setup?

There is some relevant information in Supplementary Material B, but it is difficult to understand what precisely has been assessed.

Where models have been retrained, I was unable to find full code necessary to reproduce the work.

4. A potentially significant barrier to adoption is the choice of licensing model. TIAToolbox itself is licensed under the GPL v3, a 'strong copyleft' license. However, auxiliary files (including pretrained models) are under a Creative Commons Attribution-NonCommercial-ShareAlike Version 4 (CC BY-NC-SA 4.0) license. According to the CC wiki, CC BY-NC-SA is not considered to qualify as an 'open license' under the Open Definition (https://wiki.creativecommons.org/wiki/NonCommercial_interpretation); this is in contrast to the GPL, which explicitly cannot be used to restrict commercial use (<https://www.gnu.org/licenses/gpl-faq.en.html#NoMilitary>). This creates a slightly confusing situation with code and auxiliary files under seemingly inconsistent licenses, so that potential users ought to carefully consider whether their intended use of the software (code and/or models) is permitted, in addition to what implications the licenses may have for any derived work.

I recognize that these choices may be intentional to discourage commercial use, and that a dual-licensing model is available. In my opinion, the work would be much more attractive to the wider community, including within academia, if both code and models were made available under permissive, open licenses - or at least compatible 'share-alike' terms that do not explicitly forbid commercial use (e.g. see <https://creativecommons.org/2015/10/08/cc-by-sa-4-0-now-one-way-compatible-with-gplv3/>). TIAToolbox's core dependencies are libraries that are already available more permissive open licenses, and I (as an academic researcher) would personally choose to use them directly rather than via TIAToolbox's API, because of the potential impact of licensing choices on any derived software - including open-source software.

Overall, I think TIAToolbox looks to be powerful, well designed and very well-documented. I believe that it has the potential to accelerate CPath algorithm development, and I hope that its adoption may prompt more groups to release open and reproducible code.

Minor points:

- The statement: 'because of [QuPath's] dependence on Bio-Formats, it can be difficult to integrate with a custom Python ML pipeline' is wrong; Bio-Formats is an optional dependency (QuPath can use OpenSlide and other libraries), and does not directly impact QuPath's integration with a Python ML pipeline. It is fair to say that a dependency on Java makes both QuPath and Bio-Formats awkward to use from Python, although PAQUO is a Python package that specifically addresses this (<https://pypi.org/project/paquo/>).

- The presentation of existing 'semantic segmentation' capabilities in QuPath, Fiji and Python is inaccurate; QuPath has a 'pixel classification' framework, but it is not limited to random forest classifiers and does not explicitly support variance or entropy features. The manuscript seem to conflate this QuPath-specific feature with the 'Trainable Weka Segmentation' plugin for Fiji, but these were developed independently and are completely distinct to one another in terms of codebase, features, and machine learning libraries used. 'multiscale_basic_features' from scikit-image is also an totally separate codebase, written in a different language. If the purpose is to discuss the concept of 'trainable semantic segmentation', then ilastik should be mentioned (as far as I am aware, it predates all of the above).

Reviewer #2 (Remarks to the Author):

This paper presents TIAToolbox, an open-source toolbox for image analysis in digital pathology. It is illustrated on different image analysis tasks (including patch classification, semantic segmentation, nuclear instance segmentation and classification) and provide reusable code to read whole slide images and apply a significant list of existing deep learning models.

This toolbox is written in the Python programming language and although significant software development efforts have been made to make it "usable", I believe the main target is computer/data scientists and not pathologists/end-users. Indeed, there is only a "simple web application" (line 697) while daily usage for end-users requires advanced graphical user interfaces. I am therefore a little skeptical about the claim that it is accessible to pathologists (Abstract) and about the relevance of publishing this work in a journal like Communications Medicine.

For computer scientists, this toolbox eases low-level operations (such as whole slide image reading) and the reuse of existing algorithms and pretrained AI models. Although different applications are illustrated, the toolbox will not provide a solution for all digital pathology tasks. However, it is not clear how far the toolbox could be extended (e.g. only PyTorch models are supported) and there is little explanation about how to extend the core of the library and make it scalable. While authors mention they use "easy-to-use" API (line 538), little is said about modern standards and technologies (such as OpenAPI, Docker containers, ...) that are commonly used to ease extension and reusability of a software toolbox. It is also not clear how authors will ensure reproducibility of image analysis workflows as these are relying on external libraries that also evolve over time.

Other points:

1) Table 1 and "Existing tools" Section present other software tools but some important open-source tools are missing, including:

* Cytomine (Maree et al. 2016) and its algorithm execution engine (Rubens et al., 2020)

* Orbit (<https://doi.org/10.1371/journal.pcbi.1007313>),

* ASAP (<https://computationalpathologygroup.github.io/ASAP/>)

Authors might also comment on the possibility to combine TIAToolbox with other tools rather than a competitive presentation.

2) I'm not sure that a copyleft license (GPL) will foster "commercial growth" (line 352). Various reports (e.g. "Open Source Licenses in 2022: Trends and Predictions") show that the use of permissive open source licenses continues to rise and that business-wise the preference is for licenses with fewer restrictions and limitations.

3) Authors consider their annotation store to be a potentially important contribution for the community (line 744). It would be welcome to have more details about the performance (execution time, memory requirements) of this store with hundreds of thousands or millions of annotations.

4) Similarly, there is no discussion about computational requirements and execution time to process whole slide images. There are no details about "Speed of execution" (line 513). In "Guiding principle" it is stated that a command-line interface is provided to batch-process images or WSIs on CPU/GPU clusters (line 69) but no more details/experiments are provided later in the paper.

5) Line 510: "created by Byfield": missing reference ?

Overall, I think the TIAToolbox is an important piece of work but the target audience should be clarified and additional technical details would be needed to convince data scientists to adopt and extend the tool.

Response to the Reviewers' Comments

TIAToolbox: An End-to-End Toolbox for Advanced Tissue Image Analytics

Jonathan Pocock, Simon Graham, Quoc Dang Vu, Mostafa Jahanifar, Srijay Deshpande, Giorgos Hadjigeorgiou, Adam Shephard, Raja Muhammad Saad Bashir, Mohsin Bilal, Wenqi Lu, David Epstein, Fayyaz Minhas, Nasir Rajpoot and Shan E Ahmed Raza

Dear Editor and Reviewers,

We are very grateful to the reviewers for their insightful comments and invaluable suggestions, which have resulted in significant improvement in the quality of our manuscript. Below, we give our response to each of the reviewers' comments or suggestions that needed to be addressed.

Reviewer # 1

The package looks to be very professional, with excellent documentation and test coverage.

Overall, I think TIAToolbox looks to be powerful, well designed and very well-documented. I believe that it has the potential to accelerate CPath algorithm development, and I hope that its adoption may prompt more groups to release open and reproducible code.

We thank the reviewer for their encouraging comments.

R1.1: The target audience and limitations of TIAToolbox are not clearly articulated, to the extent of giving a misleading impression of what it does. For example, the abstract describes TIAToolbox as 'a Python toolbox designed to make CPath more accessible to new and advanced CPath scientists and pathologists alike'. The inclusion of 'and pathologists alike' makes this a strong claim that I do not think is fully justified. Making use of TIAToolbox relies heavily on strong coding and data science skills...a "one-stop shop" for CPath' seems excessive for the same reasons...

A1.1: We expect the primary users of the toolbox to be CPath researchers, data scientists and developers. However, we believe that through the extensive examples and simplification of running several popular pre-trained models, this toolbox provides a starting point to clinical and biomedical researchers who are interested in testing CPath algorithms on their own data sets. We appreciate, however, that there may be a learning curve for pathologists with limited programming skills to use the toolbox.

In response to the reviewer's concerns, we have revised the abstract to

"...a Python toolbox designed to make CPath more accessible to computational, biomedical and clinical researchers".

In addition, we have modified the term "one-stop shop" in the Discussion and rephrased the relevant sentence (line 286) as follows:

"Therefore, we hope this will enable the users of TIA toolbox to access a wide and comprehensive set of tools, enabling them to focus more on model development."

R1.2: Table 1. is strongly biased in favor of showing TIAToolbox as the most comprehensive CPath solution, listing only features available in the toolbox. For example, file formats such as CZI and DICOM that are not supported by TIAToolbox are omitted, along with other potentially key features (e.g. fluorescence/multiplexed image support, interactive GUI, whole slide viewer, cross-platform installer).

A1.2: We thank the reviewer for pointing this out. We have included additional rows to Table 1, as per the reviewer's suggestions, such as including DICOM WSI, multiplexed fluorescence imaging support and the lack of an interactive GUI to the table. We have also included more tools to the table i.e., HistomicsTK and Cytomine.

R1.3: The 'Results' section focusses on two WSI prediction tasks using DL models recently proposed by members of the authorship team. This lacks clear metrics indicating the benefits for TIAToolbox; there is a reference to using 'significantly fewer lines of code', is it possible to provide numbers for this? Or perhaps performance comparisons, at least on a given hardware setup?

A1.3: We thank the reviewer for highlighting this. We have added an example in Listing A-11 to provide the metrics for the number of lines of code for a clear indication on how using our toolbox can lead to greater coding efficiency. The example implementation of patch extraction without using TIAToolbox has **16 lines** of code, and the implementation with TIAToolbox has only **5 lines** of code. Additionally, the implementation without the toolbox does not work with formats which are not supported by OpenSlide, whereas the version using the TIAToolbox extractor class supports many more WSI file types.

Figure 2a is aimed at addressing this point where most of the code in two different pipelines is implemented using TIAToolbox reducing the amount of boilerplate coding effort and duplication.

Regarding the reviewer's question about performance comparison, we would like to emphasize that we are not making any claim about performance improvements other than in the annotations. We have provided figures for performance improvements in the Annotation Store (https://github.com/TissueImageAnalytics/tiatoolbox/blob/publication/benchmarks/annotation_store.ipynb).

R1.4 There is some relevant information in Supplementary Material B, but it is difficult to understand what precisely has been assessed. Where models have been retrained, I was unable to find full code necessary to reproduce the work.

A1.4: If the reviewer is referring to reproducing results using pretrained models in the toolbox, all necessary details have already been provided in the supplementary materials B. On the other hand, if the reviewer is looking for code for reproducing the results, example Python notebooks can be found at the following URL: <https://github.com/TissueImageAnalytics/tiatoolbox/tree/publication/examples>.

The supplementary table B provides metrics for various models provided in the toolbox. If the models were retrained, the supplementary table provides metrics for the retrained model. For clarity, we have added metrics for individual models in the online documentation (readthedocs) for the toolbox. See <https://tia-toolbox.readthedocs.io/en/publication/autosummary/>

[tiatoolbox.models.architecture.micronet.MicroNet.html#tiatoolbox.models.architecture.micronet.MicroNet](https://tiatoolbox.org/models/architecture/micronet/MicroNet.html#tiatoolbox.models.architecture.micronet.MicroNet), for instance.

R1.5: A potentially significant barrier to adoption is the choice of licensing model. TIAToolbox itself is licensed under the GPL v3, a 'strong copyleft' license. However, auxiliary files (including pretrained models) are under a Creative Commons Attribution-NonCommercial-ShareAlike Version 4 (CC BY-NC-SA 4.0) license....In my opinion, the work would be much more attractive to the wider community, including within academia, if both code and models were made available under permissive, open licenses - or at least compatible 'share-alike' terms that do not explicitly forbid commercial use...TIAToolbox's core dependencies are libraries that are already available more permissive open licenses, and I (as an academic researcher) would personally choose to use them directly rather than via TIAToolbox's API, because of the potential impact of licensing choices on any derived software - including open-source software.

A1.5: In line with the reviewer's suggestion and feedback from the community, we have updated the license for the toolbox to a more open BSD 3-clause license. The pretrained models are still under CC license, as the data used for training some of these models is not available for commercial purposes.

R1.6: The statement: 'because of [QuPath's] dependence on Bio-Formats, it can be difficult to integrate with a custom Python ML pipeline' is wrong; Bio-Formats is an optional dependency (QuPath can use OpenSlide and other libraries), and does not directly impact QuPath's integration with a Python ML pipeline. It is fair to say that a dependency on Java makes both QuPath and Bio-Formats awkward to use from Python, although PAQUO is a Python package that specifically addresses this (<https://pypi.org/project/paquo/>).

A1.6: We thank the reviewer for pointing this out. We have rephrased the sentence on line 83 to:

"However, because of its dependence on **Java**, its integration with a custom Python ML pipeline may require additional steps."

R1.7: The presentation of existing 'semantic segmentation' capabilities in QuPath, Fiji and Python is inaccurate; QuPath has a 'pixel classification' framework, but it is not limited to random forest classifiers and does not explicitly support variance or entropy features. The manuscript seems to conflate this QuPath-specific feature with the 'Trainable Weka Segmentation' plugin for Fiji, but these were developed independently and are completely distinct to one another in terms of codebase, features, and machine learning libraries used. 'multiscale_basic_features' from scikit-image is also a totally separate codebase, written in a different language. If the purpose is to discuss the concept of 'trainable semantic segmentation', then ilastik should be mentioned (as far as I am aware, it predates all of the above).

A1.7: We thank the reviewer for bringing this to our attention. In line with the reviewer's comments, we have corrected the full description of QuPath's pixel classification framework, along with ilastik in the revised manuscript.

The revised text from line 94 reads:

“QuPath also provides a semantic segmentation model based on a set of classical features (such as pixel mean, variance, entropy, etc.) and a classical feature-based classifier model (such as a random forest). DL models for semantic segmentation typically produce results of higher quality, due to their ability to automatically extract representative image features. Therefore, we use cutting-edge pre-trained models in TIAToolbox which have been trained on images sampled across many slides using large public data sets, making them usable without any further user interaction or labeling. QuPath does also offer compatibility with a pre-trained StarDist deep learning model for nucleus segmentation, currently requiring manual setup by the user.”

Reviewer # 2

R2.1: This toolbox is written in the Python programming language and although significant software development efforts have been made to make it "usable", I believe the main target is computer/data scientists and not pathologists/end-users. Indeed, there is only a "simple web application" (line 697) while daily usage for end-users requires advanced graphical user interfaces. I am therefore a little skeptical about the claim that it is accessible to pathologists (Abstract) and about the relevance of publishing this work in a journal like Communications Medicine.

A2.1: We thank reviewer 2 for their comment. We have addressed the same comment in A1.1 above.

R2.2: However, it is not clear how far the toolbox could be extended (e.g. only PyTorch models are supported) and there is little explanation about how to extend the core of the library and make it scalable.

A2.2: There is no restriction on integrating models from other libraries. For example, TensorFlow models may also be integrated with the toolbox by defining a new class using `tf.keras.Model` as the base class under `TIAToolbox` models. Using a different library may require additional steps, such as exporting the model. We chose to support a single ML backend to reduce implementation/maintenance complexity and to keep installation simpler for users. PyTorch was chosen as the deep learning library for the toolbox as it supports an agile programming environment with eager execution. Additionally, it can load ONNX models exported from other frameworks, such as TensorFlow, and perform inference with them.

There is also no reason why the core of the toolbox cannot be extended by the open-source community due to its modular codebase and now more open and permissive license.

R2.3: While authors mention they use "easy-to-use" API (line 538), little is said about modern standards and technologies (such as OpenAPI, Docker containers, ...) that are commonly used to ease extension and reusability of a software toolbox.

It is also not clear how authors will ensure reproducibility of image analysis workflows as these are relying on external libraries that also evolve over time.

A2.3: The reviewer has raised a valid concern. We have now released a docker container on the GitHub Container Registry (<https://github.com/TissueImageAnalytics/tiatoolbox-docker>). In addition, we have also released a conda-forge package for the toolbox.

For reproducibility of image analysis workflows while relying on external libraries, we have set up automated unit test environments using Travis CI/CD pipelines with more than 99% unit test coverage which continuously tests the toolbox for latest *pinned* versions of dependencies.

R2.4: Table 1 and "Existing tools" Section present other software tools but some important open-source tools are missing, including:

* Cytomine (Maree et al. 2016) and its algorithm execution engine (Rubens et al., 2020)

*Orbit(<https://doi.org/10.1371/journal.pcbi.1007313>),

*ASAP(<https://computationalpathologygroup.github.io/ASAP/>)

Authors might also comment on the possibility to combine TIAToolbox with other tools rather than a competitive presentation.

A2.4: When producing Table 1, we were required to determine a cutoff for the number of compared tools. As such, we attempt to select a number of the most relevant tools and packages with a similar feature set. We have now added Histomics and Cytomine to the comparison in Table 1 as these had the most features among the libraries mentioned by the reviewer.

Tools such as QuPath, Histomics and Cytomine offer APIs which allow for integration with other tools. Via these APIs, it is possible to use features of the TIAToolbox as a backend to complement the functionality of these other tools.

R2.5: I'm not sure that a copyleft license (GPL) will foster "commercial growth" (line 352). Various reports (e.g. "Open Source Licenses in 2022: Trends and Predictions") show that the use of permissive open source licenses continues to rise and that business-wise the preference is for licenses with fewer restrictions and limitations.

A2.5: In line with the reviewer's suggestion and feedback from the community, we have updated the license for the toolbox to a more open BSD 3-clause license. The pretrained models are still under CC license, as the data used for training some of these models is not available for commercial purposes.

R2.6: Authors consider their annotation store to be a potentially important contribution for the community (line 744). It would be welcome to have more details about the performance (execution time, memory requirements) of this store with hundreds of thousands or millions of annotations.

A2.6: We thank the reviewer for their comment and have now included a benchmark notebook within the source repository to demonstrate the effectiveness of the annotation storage class in efficiently handling millions of polygon annotations.

R2.7: Similarly, there is no discussion about computational requirements and execution time to process whole slide images. There are no details about "Speed of execution" (line 513). In "Guiding principle" it is stated that a command-line interface is provided to batch-process images or WSIs on CPU/GPU clusters (line 69) but no more details/experiments are provided later in the paper.

A2.7: All the models are automatically tested on Travis CI/CD which uses 2 virtual CPU cores with 7.5 GB RAM. Therefore, these are the minimum hardware requirements for running the toolbox. The speed of execution depends on the hardware. However, we have provided execution times in Google Colab environments for the sake of transparency e.g., to process the example whole slide image for IDaRS it takes 105 seconds.

Travis: (<https://docs.travis-ci.com/user/reference/overview/#virtualisation-environment-vs-operating-system>):

- 2 virtual cores CPU
- 7.5GB RAM

Google Colab:

- 2 virtual cores from an Intel(R) Xeon(R) CPU @ 2.20GHz
- 12GB RAM
- 1/2 of a Tesla K80 (optional virtual GPU)

We would like to assert that these minimal computational requirements are not too onerous. Both Travis and Google Colab are relatively low-end hardware environments compared to a typical consumer desktop PC with an external GPU.

R2.8: Line 510: "created by Byfield": missing reference ?

A2.8: We thank the reviewer for pointing out this error and have now corrected this.

Reviewers' comments:

Reviewer #1 (Remarks to the Author):

I thank the authors for their revisions made in response to the earlier review comments. In particular, I believe that the purpose of TIAToolbox is now more clearly and accurately described, and the use of a more permissive license for the code will make this excellent work much more widely useful.

I would still encourage the authors to also consider providing the pretrained models under more permissive licenses where possible, rather than CC-BY-NC 4.0 exclusively - or at least to clarify the rationale. The statement in the rebuttal that 'The pretrained models are still under CC license, as the data used for training some of these models is not available for commercial purposes' seems inconsistent with the statement in the paper that 'Model weights are also available for commercial use on request'.

The discussion on QuPath remains somewhat confusing/inaccurate. It speaks of QuPath providing 'models' for some tasks; QuPath does not currently provide any pretrained machine learning models but rather a) conventional image processing algorithms for tasks such as cell detection, and b) tools to train and apply semantic segmentation and object classification models using conventional machine learning (e.g. random trees, ANN). As I pointed out in my previous review, QuPath does not use pixel mean, variance or entropy features as such for semantic segmentation; this appears to confuse QuPath and the (entirely separate) Trainable Weka Segmentation plugin for Fiji (<https://imagej.net/plugins/tws/>). QuPath's semantic segmentation features are included in a table at https://qupath.readthedocs.io/en/0.3/docs/tutorials/pixel_classification.html. Furthermore, QuPath's StarDist support is alluded to at the beginning of the paragraph (L92) and then explicitly mentioned at the end (L100), but in such a way that it is unclear that the references are to the same thing (and arguably with a disproportionate emphasis on the fact it involves downloading an optional extension).

Table 1 is improved, although still rather selective. For example,

- It remains the case that only file formats supported by TIAToolbox are listed. Other important formats for whole slide imaging (e.g. Zeiss CZI, Olympus VSI) that are supported by Bio-Formats (and therefore some of the other tools) are not mentioned. Bio-Formats also lists support for jp2 (see <https://docs.openmicroscopy.org/bio-formats/6.10.0/supported-formats.html>), however I am not sure if there are specific features of Omnyx images that are incompatible.

- DICOM WSI is listed as supported by TIAToolbox in the table, however the main text (L360) states only that support is planned. I understand it was a recent addition to TIAToolbox, so presume the 'planned' status in the text should be updated.

- QuPath does support patch extraction in a variety of ways, e.g.

https://qupath.readthedocs.io/en/stable/docs/advanced/exporting_images.html#tile-exporter

- It is unclear why TIAToolbox is assigned a '!' for Graphical User Interface, given that 'An exclamation mark (!) indicates a feature that may be possible with the software but either requires training or is not directly integrated with the software package'. This seems to suggest a GUI may be possible but isn't directly integrated with the software, in which case I do not understand how this differs from the other Python-based tools that were given an 'X'.

If the table is intended to show a (fairly) comprehensive list of features relevant to histopathology

image analysis, then I think functionality included in other tools - but not TIAToolbox - should be included as well, with Bio-Formats support perhaps the most clearly relevant to many potential users. In that regard, optional Bio-Formats support might be considered for a future release to extend the range of supported slide scanners, since it is achievable through Python (e.g. see <https://allencellmodeling.github.io/aicsimageio/>). However, if the table legend were to state that it provides a comparison between features in different histopathology software packages and the features included in TIAToolbox, then I think the selectivity would be more justifiable - since TIAToolbox would then explicitly be the baseline.

Reviewer #2 (Remarks to the Author):

Authors have made minor changes to the paper but they address most important comments:

- 1) The open-source license is now a permissive one so it should foster a wider use of the toolbox.
- 2) It is better clarified that primary users will be developers and data scientists
- 3) Some additional information are given (online) regarding their annotation store.
- 4) They reference additional software tools in Table 1.

Regarding this Table 1:

- Scientific references are missing for some of the tools (Histomics, Cytomine)
- I would recommend authors to contact developers to solve the "?" unclear features and platform compatibility.

Response to the Reviewers' Comments 2

TIAToolbox: An End-to-End Toolbox for Advanced Tissue Image Analytics

Jonathan Pocock, Simon Graham, Quoc Dang Vu, Mostafa Jahanifar,
Srijay Deshpande, Giorgos Hadjigeorgiou, Adam Shephard,
Raja Muhammad Saad Bashir, Mohsin Bilal, Wenqi Lu, David Epstein, Fayyaz Minhas,
Nasir Rajpoot and Shan E Ahmed Raza

Reviewer #1

R1.1: I thank the authors for their revisions made in response to the earlier review comments. In particular, I believe that the purpose of TIAToolbox is now more clearly and accurately described, and the use of a more permissive license for the code will make this excellent work much more widely useful.

A1.1: We thank the reviewer for their positive feedback.

R1.2: I would still encourage the authors to also consider providing the pretrained models under more permissive licenses where possible, rather than CC-BY-NC 4.0 exclusively - or at least to clarify the rationale. The statement in the rebuttal that 'The pretrained models are still under CC license, as the data used for training some of these models is not available for commercial purposes' seems inconsistent with the statement in the paper that 'Model weights are also available for commercial use on request'.

A1.2: If a commercial entity requires commercial use of the model weights, they can email us (email address is listed in the README file at this link <https://github.com/TissueImageAnalytics/tiatoolbox/blob/develop/README.md#dual-license>) or create an issue on the GitHub page to raise this request. Unfortunately, due to restrictions on the datasets and annotations that we have used for training, we are not allowed to permit commercial use for all the models. We also welcome community pull requests which contribute weight files with no restrictions, which may be hosted on our servers for the toolbox.

R1.3: The discussion on QuPath remains somewhat confusing/inaccurate. It speaks of QuPath providing 'models' for some tasks; QuPath does not currently provide any pretrained machine learning models but rather a) conventional image processing algorithms for tasks such as cell detection, and b) tools to train and apply semantic segmentation and object classification models using conventional machine learning (e.g. random trees, ANN). As I pointed out in my previous review, QuPath does not use pixel mean, variance or entropy features as such for semantic segmentation; this appears to confuse QuPath and the (entirely separate) Trainable Weka Segmentation plugin for Fiji (<https://imagej.net/plugins/tws/>). QuPath's semantic segmentation features are included in a table at https://qupath.readthedocs.io/en/0.3/docs/tutorials/pixel_classification.html. Furthermore, QuPath's StarDist support is alluded to at the beginning of the paragraph (L92) and then explicitly mentioned at the end (L100), but in such a way that it is unclear that the

references are to the same thing (and arguably with a disproportionate emphasis on the fact it involves downloading an optional extension).

A1.3: We have updated the manuscript as per the reviewers' suggestion on lines **91** through **107** as below:

QuPath includes some classical image processing algorithms and also integrates with some DL models as plugins. For example, it includes a semantic pixel segmentation method which utilizes a user configurable set of simple image features (e.g., color channel intensity, gradient magnitude, Laplacian of Gaussian, etc.) which are fed to specified classifiers such as a random forest, k nearest neighbors (KNN), or artificial neural network (ANN). Pre-trained DL models, for example StarDist⁸, are not included directly with QuPath but may be downloaded by a user and enabled as a plugin.

DL models typically produce results of higher quality than classical image processing methods, due to their ability to automatically extract representative image features. Therefore, we focus on including pre-trained cutting-edge pre-trained models in TIAToolbox which have been trained on images sampled across many slides using large public data sets, making them easily usable without any further user configuration or labelling.

Other Python software packages, such as PathML, offer some trained deep learning models. However, the selection is often limited, currently only one model (HoVer-Net) in the case of Dana-Farber-AIOS PathML, with a U-Net implementation in progress. There is also no clearly documented way to integrate additional models or custom user models with PathML.

R1.4: Table 1 is improved, although still rather selective. For example,
- It remains the case that only file formats supported by TIAToolbox are listed. Other important formats for whole slide imaging (e.g. Zeiss CZI, Olympus VSI) that are supported by Bio-Formats (and therefore some of the other tools) are not mentioned.

A1.4: We already included a row in Table 1 indicating general support for fluorescence imaging. However, we have now added rows explicitly for CZI and VSI support. TIAToolbox does have limited support for reading fluorescence images in the TIFF format. We recognize that this is an important feature, which we aim to address in future work with support for Zeiss CZI and PerkinElmer QPTIFF formats.

R1.5: Bio-Formats also lists support for jp2 (see <https://docs.openmicroscopy.org/bio-formats/6.10.0/supported-formats.html>), however I am not sure if there are specific features of Omnyx images that are incompatible.

A1.5: To the best of our knowledge, Bio-Formats does support JP2 in general, but not Omnyx JP2 in particular. It can handle small visual fields in the JP2 format. However, it does not fully support reading WSIs in the JP2 file format.

R1.6: DICOM WSI is listed as supported by TIAToolbox in the table, however the main text (L360) states only that support is planned. I understand it was a recent addition to TIAToolbox, so presume the 'planned' status in the text should be updated.

A1.6: We thank the reviewer for highlighting this mistake. We have corrected this in the manuscript on line 360. This is updated on lines **358** through **359**. Additionally, we have added support for NGFF 4.0 with CYX (channel, height, width) axes. The revised text reads:

Lastly, we include support for reading WSI DICOM images (via wsidicom) with JPEG and JPEG2000 compressed tiles. Furthermore, we include experimental support for a developing next generation file format (NGFF version 0.4) based on Zarr²³.

R1.7: QuPath does support patch extraction in a variety of ways, e.g.

https://gupath.readthedocs.io/en/stable/docs/advanced/exporting_images.html#tile-exporter

A1.7: According to the link shared by the reviewer, QuPath does indeed support patch extraction and export of the images. We have amended Table 1 to reflect QuPath support for patch extraction. However, TIAToolbox additionally supports patch extraction through a Python lazy (data on demand) iterable as highlighted on lines **443** to **445**. This is more memory efficient for streaming deep learning pipelines rather than writing all patches to disk first. Saving these images to disk may require a large amount of space if a lossless encoding is used to avoid re-encoding artifacts.

R1.8: It is unclear why TIAToolbox is assigned a '!' for Graphical User Interface, given that 'An exclamation mark (!) indicates a feature that may be possible with the software but either requires training or is not directly integrated with the software package'. This seems to suggest a GUI may be possible but isn't directly integrated with the software, in which case I do not understand how this differs from the other Python-based tools that were given an 'X'.

A1.8: TIAToolbox does provide a tool to interactively display whole slide images via a web browser. However, this does not provide the full functionality of displaying or editing annotations and other interactive functionality. Therefore, we have indicated partial implementation. We have clarified the exclamation mark for partial implementation in the manuscript in Table 1 caption on lines **157** through **160**. The relevant edited caption sections now read:

... An exclamation mark (!) indicates a feature that may be partially implemented or is possible with the software, but either requires training or is not directly integrated with the software package. ...

... ¹TIAToolbox includes a basic web based UI for displaying WSIs on top of each other with adjustable opacity, but not a full-featured GUI.

R1.9: If the table is intended to show a (fairly) comprehensive list of features relevant to histopathology image analysis, then I think functionality included in other tools - but not TIAToolbox - should be included as well, with Bio-Formats support perhaps the most clearly relevant to many potential users. In that regard, optional Bio-Formats support might be considered for a future release to extend the range of supported slide scanners, since it is

achievable through Python (e.g. see <https://allencellmodeling.github.io/aicsimageio/>). However, if the table legend were to state that it provides a comparison between features in different histopathology software packages and the features included in TIAToolbox, then I think the selectivity would be more justifiable - since TIAToolbox would then explicitly be the baseline.

A1.9: TIAToolbox does not currently support Bio-Formats due to the GPL licensing of Bio-Formats (and python bioformats), the requirement for javabridge and increased complexity with cross-platform support during installation of the toolbox.

However, Bio-Formats functionality can be set up by adding a subclass of WSISReader in a fork of the toolbox if required. We show an example proof-of-concept implementation of how this can be done in a GitHub gist: <https://gist.github.com/John-P/7da4224d76eedfd6a5f6239589acaf2f>. In the future, we expect this to be an optional extension package, which can be imported in addition to the main toolbox to allow use of Bio-Formats with TIAToolbox.

Reviewer #2

R2.1: Authors have made minor changes to the paper but they address most important comments:

- 1) The open-source license is now a permissive one so it should foster a wider use of the toolbox.
- 2) It is better clarified that primary users will be developers and data scientists
- 3) Some additional information are given (online) regarding their annotation store.
- 4) They reference additional software tools in Table 1.

A2.1: We thank the reviewer for their positive feedback.

R2.2: Regarding this Table 1:

- Scientific references are missing for some of the tools (Histomics, Cytomine)

We thank the reviewer for bringing this to our attention. We have addressed this in the manuscript.

- I would recommend authors to contact developers to solve the "?" unclear features and platform compatibility.

A2.2: Indeed, the Table was compiled after consultation with the developers of Histomics and Cytomine. We have used the labels in the table which they suggested reflecting the current feature set.

REVIEWERS' COMMENTS:

Reviewer #1 (Remarks to the Author):

I respect the authors' right and decision to choose their licensing model for model weights, but continue to find it inconsistent to justify this by 'we are not allowed to permit commercial use for all the models' while retaining the statement in the paper that 'Model weights are also available for commercial use on request'.

Trying to reconcile this, I understand the statement could mean either 'some model weights' (in which case their licensing remains the authors' choice) or 'different model weights' (generated without using certain training data, resulting in different weights). Which meaning is unclear.

Apart from this, I thank the authors for making the other changes and congratulate them on their excellent work.

Response to the Reviewers' Comments

TIAToolbox: An End-to-End Toolbox for Advanced Tissue Image Analytics

Jonathan Pocock, Simon Graham, Quoc Dang Vu, Mostafa Jahanifar,
Srijay Deshpande, Giorgos Hadjigeorgiou, Adam Shephard,
Raja Muhammad Saad Bashir, Mohsin Bilal, Wenqi Lu, David Epstein, Fayyaz Minhas,
Nasir Rajpoot and Shan E Ahmed Raza

Reviewer #1

R1.1: I respect the authors' right and decision to choose their licensing model for model weights, but continue to find it inconsistent to justify this by 'we are not allowed to permit commercial use for all the models' while retaining the statement in the paper that 'Model weights are also available for commercial use on request'.

We thank the reviewers for their comments. We have updated the licencing information on lines 678-679 as follows: "Model weights can be made available for commercial use on request depending on ethical approvals from the data source."